# Least squares variational inference

**Yvann Le Fay**[1], **Nicolas Chopin**[1,] *, **Simon Barthelmé**[2]
[1] ENSAE, CREST, IP Paris
[2] GIPSA-Lab, CNRS
{yvann.lefay,nicolas.chopin}@ensae.fr
simon.barthelme@gipsa-lab.fr

## Abstract

Variational inference seeks the best approximation of a target distribution within a chosen family, where "best" means minimising Kullback-Leibler divergence. When the approximation family is exponential, the optimal approximation satisfies a fixed-point equation. We introduce LSVI (Least Squares Variational Inference), a gradient-free, Monte Carlo-based scheme for the fixed-point recursion, where each iteration boils down to performing ordinary least squares regression on tempered log-target evaluations under the variational approximation. We show that LSVI is equivalent to biased stochastic natural gradient descent and use this to derive convergence rates with respect to the numbers of samples and iterations. When the approximation family is Gaussian, LSVI involves inverting the Fisher information matrix, whose size grows quadratically with dimension $d$. We exploit the regression formulation to eliminate the need for this inversion, yielding $\mathcal{O}(d^3)$ complexity in the full-covariance case and $\mathcal{O}(d)$ in the mean-field case. Finally, we numerically demonstrate LSVI's performance on various tasks, including logistic regression, discrete variable selection, and Bayesian synthetic likelihood, showing results competitive with state-of-the-art methods, even when gradients are unavailable.

## 1 Introduction

This paper focuses on parametric variational inference (VI, [1–3]). Given an (unnormalised) target density $\pi$, we aim at finding the distribution that minimises the (reverse) Kullback-Leibler divergence:

$$\arg \min_{q \in \mathcal{Q}} \mathrm{KL}\left(q \mid \bar{\pi}\right) \coloneqq \int q \log\left(q/\bar{\pi}\right) \tag{1}$$

where $\mathcal{Q}$ is a user-chosen parametric family (e.g., Gaussians), and $\bar{\pi} = \pi / \int \pi$. This approach has become a de facto standard in probabilistic machine learning in recent years and is implemented in various software packages, such as STAN, NumPyro, PyMC3, and Blackjax [4–7]. The minimisation is typically carried out through gradient-based procedures using automatic differentiation, either stochastic gradient descent (SGD, [8])—often applied after reparameterising the target distribution [9, 10]—or its faster alternative natural gradient descent (NGD, [11–14]). This is convenient for users, as they only have to provide the function $f \coloneqq \log \pi$ to the software.

These procedures use different gradient estimators; some require $\log \pi$ to be amenable to automatic differentiation, which is the case when using a reparameterisation, while others only require gradient estimators of expectations under the variational distribution via the log-derivative trick [15]. The gradient estimator for expectations usually suffers from high variance, and practical implementations

---

*Corresponding author

39th Conference on Neural Information Processing Systems (NeurIPS 2025).

rely on the reparameterisation trick, which is not possible in several important cases, for instance when $\pi$ is a discrete distribution, or when $\pi$ is intractable or non-differentiable (as in likelihood-free inference). Additionally, convergence of SGD is sometimes slow and/or tedious to assess, and requires careful step sizes tuning [16] while a naive implementation of NGD requires costly matrix inversions.

## 1.1 Outline and contributions

We introduce practical algorithms for VI within exponential families when gradients of $\log \pi$ are unavailable. These algorithms involve taking biased stochastic gradient descent steps, but we show both theoretical convergence and good performance in non-toy problems. In Section 2, we derive an exact, but intractable, iteration we call LSVI, that boils down to performing successive least squares (OLS) regression. We highlight connections to NGD and discuss its convergence properties. In Section 3, we introduce a stochastic variant that updates the OLS estimate using multiple draws from the current approximation. Importantly, under standard smoothness and relative convexity assumptions on the objective, and bounded-moment assumptions on the variational family, we establish convergence guarantees and rates with respect to the numbers of draws and iterations, conditioned on high-probability events. In addition, we provide an adaptive method to calibrate step sizes by controlling the linear regression residuals. Section 4 focuses on the Gaussian variational family; we propose a reparametrisation of the linear regression such that the OLS procedure requires no inversion of the Fisher information matrix (FIM). These schemes tailored to Gaussian distributions are cost-efficient: our methods scale linearly with $d$ in the mean-field case, and in the full-covariance case, the cost matches the cost of computing $d \times d$ matrix products, i.e., $\mathcal{O}(d^3)$. In Section 5, we extensively illustrate the performance of our methods compared to other inference procedures, including gradient-based VI and exact Bayesian inference procedures. Limitations are discussed in Section 6. We provide a Python package supporting GPU parallelisation via JAX to replicate the experiments: `https://github.com/ylefay/LSVI`.

## 2 Exact LSVI

Let $\pi : \mathcal{X} \to \mathbb{R}$ be some unnormalised target density, with $\mathcal{X} \subset \mathbb{R}^d$. It will be convenient to work with an exponential family $\mathcal{Q}$ of *unnormalised* densities:

$$q_\eta(x) := \exp\{\eta^\top s(x)\}, \qquad \eta \in \mathcal{V} := \{\eta : Z(q_\eta) < \infty\}$$

where $\eta \in V$ is the natural parameter associated to $q_\eta \in \mathcal{Q}$, $Z(q) := \int_\mathcal{X} q$ denotes the partition function, and $s : \mathcal{X} \to \mathbb{R}^m$ the extended statistic function defined as

$$s(x) = \begin{pmatrix} 1 \\ \bar{s}(x) \end{pmatrix}, \quad \bar{s} : \mathcal{X} \to \mathbb{R}^{m-1}.$$

In words, we include an intercept in $s$ to make the family closed under multiplication by a positive scalar. For $\eta = \left(\eta^{(0)}, \bar{\eta}^\top\right)^\top$, where $\eta^{(0)}$ denotes the first component of $\eta$, let $\bar{q}_{\bar{\eta}}$ be the normalised version of $q_\eta$ (which therefore depends only on $\bar{\eta}$): $\bar{q}_{\bar{\eta}} = q_\eta/Z_\eta$, using the short-hand $Z_\eta := Z(q_\eta)$. Notation $\mathbb{E}_\eta[\cdot]$ means a properly normalised expectation, i.e. $\mathbb{E}_\eta[h] = \int_\mathcal{X} q_\eta h / Z_\eta$. Likewise, we replace the standard Kullback-Leibler objective with a divergence for unnormalised densities [17], which is defined by

$$\mathrm{uKL}(q \mid \pi) := \int q \log\left(\frac{q}{\pi}\right) + Z(\pi) - Z(q), \tag{2}$$

for any density $q$ absolutely continuous with respect to $\pi$. In addition, we assume the variational family $\mathcal{Q}$ is minimal and regular, which is a standard assumption in VI [12, 18–20], and is met by any standard exponential families (e.g., Gaussian, Beta, Poisson, Bernoulli, etc., [21, Table 3.1]). These assumptions ensure $\eta \in \mathcal{V} \mapsto q_\eta$ is injective and the log-partition function is differentiable everywhere [21, Prop. 3.1].

**Assumption 2.1** (Minimality and regularity of $\mathcal{Q}$)**.** The components of $s$ are linearly independent (minimality), and the set of natural parameters $\mathcal{V}$ is open (regularity).

The next proposition shows that the critical points of the uKL divergence are also critical points of the KL divergence, and vice versa. In words, nothing is lost by considering the uKL instead of KL.

**Proposition 2.2.** *Let $\eta = (\eta^{(0)}, \bar{\eta}^\top)^\top \in \mathcal{V}$, if $\nabla_\eta \operatorname{uKL}(q_\eta \mid \pi) = 0$ then $\nabla_{\bar{\eta}} \operatorname{KL}(\bar{q}_{\bar{\eta}} \mid \bar{\pi}) = 0$, and the reciprocal holds: $\nabla_{\bar{\eta}} \operatorname{KL}(\bar{q}_{\bar{\eta}} \mid \bar{\pi}) = 0$ and $\partial_{\eta^{(0)}} \operatorname{uKL}(q_\eta \mid \pi) = 0$, then $\nabla_\eta \operatorname{uKL}(q_\eta \mid \pi) = 0$.*

The first-order condition of the uKL minimisation problem is given by the following proposition.

**Proposition 2.3.** *Let $f = \log \pi$ be the (unnormalised) log target density. Let $\eta = (\eta^{(0)}, \bar{\eta}^\top)^\top \in \mathcal{V}$, $\nabla_\eta \operatorname{uKL}(q_\eta \mid \pi) = 0$ if and only if $\left\{ \mathbb{E}_\eta[ss^\top] \right\} \eta = \mathbb{E}_\eta[fs]$. Furthermore, if $\nabla_\eta \operatorname{uKL}(q_\eta \mid \pi) = 0$, then $\eta^{(0)} = -\operatorname{KL}(\bar{q}_{\bar{\eta}} \mid \bar{\pi}) + \log \left( Z(\pi) / \int_{\mathcal{X}} \exp \left( \bar{\eta}^\top \bar{s} \right) \right).$*

## 2.1 The exact LSVI scheme

The first-order optimality condition is equivalent to the fixed point equation: $\eta = \phi(\eta)$ with

$$\phi(\eta) := F_\eta^{-1} z_\eta, \quad F_\eta := \mathbb{E}_\eta[ss^\top], \quad z_\eta := \mathbb{E}_\eta[fs], \tag{3}$$

and $F_\eta$ is the Fisher information matrix (FIM) associated to $q_\eta$. Salimans and Knowles [22] remark that $\phi$ is the ordinary least squares regressor (OLS; [23]) of $f(X)$ with respect to $s(X)$ when $X \sim q_\eta$:

$$\phi(\eta) = \operatorname{argmin}_{\beta \in \mathbb{R}^m} \mathbb{E}_\eta \left[ \left\{ \beta^\top s(X) - f(X) \right\}^2 \right]. \tag{4}$$

A nice property of $\phi$ when $\pi$ is in the variational family with $\pi = q_{\eta^\star}$, is that for any $\eta \in \mathcal{V}$, $\phi(\eta) = \eta^\star$, i.e., $\phi$ exactly recovers $\pi$. However, in general $\phi(\eta)$ may not be in $\mathcal{V}$, and naively performing a fixed-point scheme can lead to unstable variational approximations or, worse, result in non-normalisable densities (i.e., $\phi(\eta) \notin \mathcal{V}$). To address this, we consider a relaxation of the fixed-point scheme obtained via a momentum fixed-point iteration [24]:

$$\eta_{t+1} := \varepsilon_t \phi(\eta_t) + (1 - \varepsilon_t)\eta_t, \qquad t \geq 0 \tag{5}$$

where $\varepsilon_t > 0$ is such that $\eta_{t+1}$ is in $\mathcal{V}$. Such an $\epsilon_t$ necessarily exists because $\mathcal{V}$ is open (Assumption 2.1). Since iteration (5) assumes that one has access to expectations under the variational family (which in general is not the case), we refer to (5) as the *exact* Least Squares Variational Inference (LSVI) iteration. This relaxation has a natural interpretation in this specific context: $\eta_{t+1}$ in (5) is the solution of the least squares objective (4) when $\pi = \exp f$ is replaced by the tempered (annealed) density $q_{\eta_t}^{1-\varepsilon_t} \pi^{\varepsilon_t}$.

## 2.2 LSVI as natural gradient descent (NGD) and mirror descent (MD)

This subsection summarises a well-established connection between NGD and MD in the variational inference literature [12, 18, 20] but generalised to the *unnormalised* KL divergence.

Let us define the (unnormalised) moment parameter mapping $\omega : \eta \in \mathcal{V} \mapsto \nabla_\eta Z_\eta = \int s(x) q_\eta(x)$, and let $\mathcal{W} = \omega(\mathcal{V})$ be the set of moment parameters. We denote by $\eta : \mathcal{W} \to \mathcal{V}$ the inverse mapping of $\omega : \mathcal{W} \to \mathcal{V}$, whose existence is guaranteed under Assumption 2.1 [21, Ch. 3]. Define $l$ as the unnormalised KL divergence (2). When expressed in natural parameters, we write $l : \eta \in \mathcal{V} \mapsto \operatorname{uKL}(q_\eta \mid \pi)$, when expressed in moment parameters, we write $l : \omega \in \mathcal{W} \mapsto \operatorname{uKL}(q_\omega \mid \pi)$, and similarly for expectations: $\mathbb{E}_\omega := \mathbb{E}_{\eta(\omega)}$.

The following proposition states that LSVI iteration (5) is a NGD iteration on the uKL divergence in the natural space of parameters, and equivalently a MD in the moment space [25, 26, Ch. 3].

**Proposition 2.4** (LSVI is NGD which is equivalent to MD, [20, Lemma 1]). *Under Assumption 2.1 and provided the sequence $(\eta_t)$ defined by (5) is in $\mathcal{V}$, $(\eta_t)$ satisfies the dynamic (NGD),*

$$\eta_{t+1} = \eta_t - \varepsilon_t F_{\eta_t}^{-1} \nabla_\eta l(\eta_t)/Z_{\eta_t}, \tag{6}$$

*or equivalently,*

$$\eta_{t+1} = \eta_t - \varepsilon_t \nabla_\omega l(\omega(\eta_t)). \tag{7}$$

*Furthermore, let $\omega_0 \in \mathcal{W}$ and define for $t \geq 0$ (MD),*

$$\omega_{t+1} := \operatorname{argmin}_{\omega \in \mathcal{W}} \left\{ \nabla_\omega^\top l(\omega_t)\omega + \varepsilon_t^{-1} D_{Z^*}(\omega, \omega_t) \right\}, \tag{8}$$

*where $D_{Z^\star}$ is the Bregman divergence [27] with respect to $Z^*$ the Legendre transform of $Z$: $Z^*(\omega) = \operatorname{argmin}_{\eta \in \mathcal{V}} \{\eta^\top \omega - Z_\eta\}$. Then, the sequence $(\eta_t)$ defined by (5) with $\eta_0 = \eta(\omega_0)$ satisfies for all $t \geq 0$, $\eta_t = \eta(\omega_t)$. In words, LSVI performs a natural gradient step in the space of natural parameters, which corresponds to a mirror descent step in the dual (moment) space.*

---

**Algorithm 1** Generic LSVI (any family $\mathcal{Q}$)

---

**Require:** $\eta_0 \in \mathcal{V}$, $N \geq 1$
    $\hat{\eta}_0 \leftarrow \eta_0$
    **while** not converged **do**
        $X_1, \ldots, X_N \sim q_{\hat{\eta}_t}$
        $\hat{F}_{\hat{\eta}_t} \leftarrow \frac{1}{N} \sum_{i=1}^{N} s(X_i) s^\top(X_i)$
        $\hat{z}_{\hat{\eta}_t} \leftarrow \frac{1}{N} \sum_{i=1}^{N} s(X_i) f(X_i)$
        $\hat{\eta}'_{t+1} \leftarrow \hat{F}_{\hat{\eta}_t}^{-1} \hat{z}_{\hat{\eta}_t}$                              ▷ ordinary least squares estimator (OLS)
        $\varepsilon_t \leftarrow \texttt{stepsize}(\hat{F}_{\hat{\eta}_t}, \hat{z}_{\hat{\eta}_t}, \hat{\eta}'_{t+1}, \hat{\eta}_t, X)$
        $\hat{\eta}_{t+1} \leftarrow \varepsilon_t \hat{\eta}'_{t+1} + (1 - \varepsilon_t) \hat{\eta}_t$
    **end while**

---

Proposition 2.4 allows us to leverage known convergence results for MD under standard smoothness and convexity assumptions on the uKL objective [in the VI literature, see, 12, 28, 29].

**Assumption 2.5.** The uKL objective $l : \omega \in \mathcal{W} \mapsto \text{uKL}(q_\omega \mid \pi)$ is $L$-smooth, $\mu$-strongly convex relative to $D_{Z^*}$.

Under Assumption 2.5, MD is known to converge with rate $O(1/k)$ for sufficiently small and linearly decreasing step sizes: $\varepsilon_t = (L + \alpha t)^{-1}$ for $0 \leq \alpha < \mu$, [see, e.g., 30, 31, Theorem 4.5 and Lemma 4.8, Theorem 4]. The non-strongly convex case ($\mu = 0$) exhibits $O(1/\sqrt{k})$ convergence rate for a specific choice of step sizes [see, e.g., 30, Corollary 4.6]. In practice, it is not trivial to set the $\varepsilon$ to obtain a $O(1/k)$ rate, as the relative strong convexity parameter $\mu$, if it exists, might be unknown and eventually very small.

*Remark* 2.6. The strong-convexity/convexity assumptions are rarely verified in practice, however, such assumptions are standard for analysing convergence of optimisation algorithms (including NGD and MD), to ensure a unique minimiser and tractable rates [see, e.g., 32, Ch. 5]. While non-conjugate VI objectives may not be globally convex [20] (but it holds when the variational family contains the target), *local* convexity near optima often suffices for local convergence to hold.

See [33, 34] for provable smoothness guarantees on the KL objective.

## 3 Practical algorithms and their analysis

The exact LSVI mapping $\phi$ assumed that one has access to expectations under the variational family. In practice, exact computation of those expectations is intractable for a general target log-density $f$. In this section, we introduce a practical algorithm in which these expectations are estimated via Monte Carlo, and we study the impact of the Monte Carlo error on the convergence guarantees.

### 3.1 Generic LSVI

Our first algorithm comes down to replacing the two expectations in (3) with Monte Carlo estimates:

$$\hat{F}_\eta := \frac{1}{N} \sum_{i=1}^{N} s(X_i) s(X_i)^\top, \quad \hat{z}_\eta := \frac{1}{N} \sum_{i=1}^{N} f(X_i) s(X_i), \tag{9}$$

where $X_1, \ldots, X_N \overset{\text{i.i.d.}}{\sim} q_\eta$. The counterpart to the exact iteration (5) is then

$$\hat{\eta}_{t+1} := \varepsilon_t \hat{F}_{\hat{\eta}_t}^{-1} \hat{z}_{\hat{\eta}_t} + (1 - \varepsilon_t) \hat{\eta}_t, \tag{10}$$

with $\hat{\eta}_0 = \eta_0$. At any iteration $t \geq 1$, the step size $\varepsilon_t$ can, in all generality, depend on the current state of the algorithm via a function $\texttt{stepsize}$. We discuss one possible choice in Section 3.2. This leads naturally to generic LSVI Algorithm 1, whose one-iteration cost is $\mathcal{O}(m^3 + m^2 N)$.

Iteration (10) replaces the exact computation of $F_\eta^{-1}$ with a Monte Carlo estimate $\hat{F}_\eta^{-1}$. This approximation introduces a bias in the estimation of the inverse FIM, and consequently, in the estimation of the natural gradient involved in (6). Further analysis of the statistical properties of the sequence

$(\hat{\eta}_t)$, in particular, its convergence toward a neighbourhood of the optimum, requires a careful control of the bias. When $s$ admits uniformly bounded fourth-order moment, and the spectrum of $F_\eta$ is bounded away from zero, the bias conditioned to a high-probability event can be controlled.

**Assumption 3.1.** The sufficient statistic $s$ admits uniformly bounded fourth-order moments:

$$\mu_4 := \sup_{\omega \in \mathcal{W}} \max_{1 \leq i \leq m} \left( \mathbb{E}_\omega \left[ |s(X)_i|^4 \right] \right)^{1/4} < \infty, \quad \nu := \sup_{\omega \in \mathcal{W}} \sup_{\|u\|=1, u \in \mathbb{R}^m} \left( \mathbb{E}_\omega \left[ |u^\top s(X)|^4 \right] \right)^{1/4} < \infty. \tag{11}$$

**Assumption 3.2.** The smallest spectral value $r := \inf_{\omega \in \mathcal{W}} \|F_\omega^{-1}\|^{-1}$ is strictly positive.

Both assumptions are verified if i) $\mathcal{W}$ is a compact set, and ii) $s(X)$ admits fourth-order moments, for $X \sim q_\omega$ and for all $\omega \in \mathcal{W}$. While $\mathcal{W}$ is generally not a compact set, it should not be considered as a limiting assumption in practice, and can be lifted, see [35–37]. We further assume that $f$ admits uniformly bounded second-order moment as this is required to control the norm of $\hat{z}_\omega$.

**Assumption 3.3.** $m_2 := \sup_{\omega \in \mathcal{W}} \mathbb{E}_\omega[f^2]^{1/2} < \infty$.

We derive the convergence in expectation to the minimum of the KL loss conditioned on the event that the estimated FIMs are well-conditioned.

**Theorem 3.4** (Explicit convergence rates for LSVI). *Assume 2.1, 2.5, 3.1, 3.2, and 3.3. Let $k \geq 0$, and let $\hat{\eta}_0, \hat{\eta}_1, \ldots, \hat{\eta}_k$ be given by (10), with $\hat{\omega}_t = \omega(\hat{\eta}_t)$ for $0 \leq t \leq k$. Let $\mathcal{A}_k = \cap_{t=0}^k \mathcal{A}(\hat{\omega}_t)$ with $\mathcal{A}(\omega) = [\|F_\omega - \hat{F}_\omega\| < \|F_\omega^{-1}\|^{-1}]$. Further assume that at each iteration $t \geq 1$, the quantities $\hat{F}_{\hat{\eta}_t}$ and $\hat{z}_{\hat{\eta}_t}$ are computed using two independent sets of samples. Let $c_t = c_{t-1}\varepsilon_{t-1}^{-1}(\varepsilon_t^{-1} - \mu)^{-1}$ for $t \geq 1$, $c_0 = 1$, $C_k = \sum_{t=1}^k c_{t-1}$. Let $\bar{\omega}_k = \frac{1}{C_k} \sum_{t=1}^k c_{t-1}\hat{\omega}_t$ be the weighted average of the iterates, and let $\omega^*$ be the minimiser of $l$.*

1. *Fix $\delta \in (0, 1)$, provided $\sqrt{N} \geq C_0 r^{-2}(k+1)\delta^{-1}(\sqrt{\log(m)}\mu_4\nu + \mu_4^2 \log(m))$ for some constant $C_0 > 0$, $\mathcal{A}_k$ happens with probability at least $1 - \delta$.*

2. *Conditioned on $\mathcal{A}_k$,*

$$\mathbb{E}[l(\bar{\omega}_k) \mid \mathcal{A}_k] - l(\omega^*) \leq \frac{(\varepsilon_0^{-1} - \mu)\, \mathrm{uKL}(q_{\omega^*} \mid q_{\omega_0})}{C_k} + \mathcal{O}\left(\frac{1}{N}\right) \sum_{t=0}^{k-1} \frac{c_t \varepsilon_t}{C_k} + \mathcal{O}\left(\frac{1}{N}\right), \tag{12}$$

   *where the big-$\mathcal{O}$ terms are independent of $k$.*

3. *Let $\varepsilon_t^{-1} = L + \alpha t$ for some $\alpha > 0$. The RHS of (12) has asymptotic convergence rates that depend on $\alpha$ compared to the strong-convexity parameter $\mu$. When $\alpha > \mu$, the sequence $(c_t)$ is strictly decreasing, and the rate is $\mathcal{O}\left(k^{-\mu/\alpha}\right) + \mathcal{O}\left(N^{-1}\right)$. When $\alpha = \mu$, the sequence $(c_t)$ is constant, and the rate is $\mathcal{O}(k^{-1}) + \mathcal{O}\left(\log(k)k^{-1}N^{-1}\right) + \mathcal{O}\left(N^{-1}\right)$. When $\alpha < \mu$, the sequence $(c_t)$ is strictly increasing, and the rate is $\mathcal{O}\left(k^{-\mu/\alpha}\right) + \mathcal{O}\left(k^{-1}N^{-1}\right) + \mathcal{O}\left(N^{-1}\right)$.*

*Remark* 3.5. Our proof follows a similar strategy to that of Hanzely and Richtárik [30], extending their mirror descent lemma to biased estimates. We control both the bias and the variance of the FIM estimate, conditionally on the event that the estimated FIM is well-conditioned ($\mathcal{A}_k$). We show this occurs with high probability when $N$ is sufficiently large, using concentration inequalities for positive-definite matrices [38].

*Remark* 3.6. The convergence guarantees can be decomposed in three terms. The first term is due to initialisation and vanishes as $k \to \infty$, the third term is the Monte Carlo bias and vanishes as $N \to \infty$, and the second is a cross term and vanishes whenever $k \to \infty$ or $N \to \infty$.

*Remark* 3.7. The OLS estimate to the regression problem uses a single set of samples to compute both $\hat{F}_{\hat{\eta}_t}$ and $\hat{z}_{\hat{\eta}_t}$, contrary to the estimate introduced in the previous theorem. Additionally, for many exponential families, closed-form expressions for $F_\eta$ are known. Since the OLS is optimal with respect to the variance, it exhibits lower variance compared to others estimates. Importantly, it is inefficient to use two distinct set of samples or to replace the estimated FIM with the exact FIM [22].

## 3.2 The choice of the $\varepsilon_t$'s

Setting $\varepsilon$ to a small enough and linearly decreasing sequence of step sizes ensures convergence of the sequence (10) to a neighbourhood of a local minimizer $\eta^\star$ [30, 31], see Theorem 3.4. However, the smoothness and strong-convexity parameters $(L, \mu)$ of the KL objective, if they exist, are rarely known in practice [33, 34]. For these reasons, choosing the $\varepsilon_t$ can be a tedious task, as in any stochastic optimisation scheme [16]: step sizes that are too large lead to unstable behaviours while too small step sizes lead to slow convergence.

Let $\eta \in \mathcal{V}$ and $\eta^\star = \phi(\eta)$ be the OLS, consider the following linear regression objective,

$$f(X_i) = \eta^{\star\top} s(X_i) + v_i, \quad X_1, \ldots, X_N \overset{\text{i.i.d.}}{\sim} q_\eta, \tag{13}$$

where $v_i$ is the residual of the regression. Then (13) implies that for any $\varepsilon \in (0, 1]$

$$\varepsilon f(X_i) + (1 - \varepsilon)\eta^\top s(X_i) = (\varepsilon\phi(\eta) + (1 - \varepsilon)\eta)^\top s(X_i) + \varepsilon v_i. \tag{14}$$

The previous equation (14) shows that descending toward the direction of the OLS with step size $\varepsilon$ multiplies the variance of the residuals $v_1, \ldots, v_N$, $v^2$ by $\varepsilon^2$. Let $u^2$ be some upper bound on the variance of the residuals, and let $\varepsilon \leq u/v$, then the residuals have variance less than $u^2$. This remark, combined with a backtracking procedure to ensure that the iterates remain in the set of natural parameters, yields an adaptive schedule for choosing the step sizes (Algorithm 4 in Appendix B), which we have found to be robust against noisy iterates and slow descents.

## 4 Gaussian families

The two most commonly-used families $\mathcal{Q}$ in variational inference are the full-covariance Gaussian family ($N_d(\mu, \Sigma)$ with arbitrary $\mu$ and $\Sigma \succ 0$), and the mean-field Gaussian family ($\Sigma$ is diagonal). A single iteration of LSVI requires inverting the Fisher information matrix (FIM) $F$, which is too expensive to be practical in high-dimension; i.e., $\mathcal{O}(m^3)$, with $m = \mathcal{O}(d)$ (resp. $m = \mathcal{O}(d^2)$) in the mean-field (resp. full-covariance) case.

Attempts to lessen the computational complexity of inference procedures over Gaussian distributions either rely on access to cheap gradient estimates in the space of moments [13, 18, 20, 39], on single draw updates making the FIM estimate cheap to invert but noisy [13, 40], or on restrictive assumptions on the target density [41, 42]. We derive closed-form formulae for the natural gradient descent iteration whose cost, in the full-covariance case, essentially amounts to the cost of computing $d \times d$ matrix products, that is $\mathcal{O}(d^3)$. In the mean-field case, the cost is $\mathcal{O}(d)$.

**Full-covariance Gaussian family** Let $\mathcal{Q}$ be the family of (unnormalised) Gaussian densities of dimension $d$. The sufficient statistic is $s(x) := (1, x^\top, (\text{vec}\,(xx^\top))^\top)^\top \in \mathbb{R}^m$ with $m = d(d+1) + 1$, where $\text{vec}\,(xx^\top)$ denotes the vector obtained by vertically stacking the columns of $xx^\top$, and we denote by unvec the inverse operation. Consider a natural parameter $\eta = (\eta^{(0)}, \eta^{(1),\top}, \eta^{(2),\top})^\top \in \mathcal{V}$ with $\eta^{(0)} \in \mathbb{R}$, $\eta^{(1)} \in \mathbb{R}^d$ and $\eta^{(2)} \in \mathbb{R}^{d^2}$, then it defines a unique Gaussian distribution with mean and covariance matrix given by

$$(\mu, \Sigma) = \left(-\frac{1}{2}\eta^{(2),-1}\eta^{(1)}, -\frac{1}{2}\,\text{unvec}(\eta^{(2)})^{-1}\right). \tag{15}$$

We reparameterise the linear regression of $f(X)$ with respect to $s(X)$, where $X \sim N(\mu, \Sigma)$, into a regression of $f(\mu + CZ)$ with respect to $t(Z)$, where $Z \sim N(0, I_d)$, and $C = \text{Chol}(\Sigma)$ is the Cholesky of $\Sigma$, and

$$t(z) := \left(1, z^\top, \frac{z_1^2 - 1}{\sqrt{2}}, z_1 z_2, \ldots, z_1 z_d, \frac{z_2^2 - 1}{\sqrt{2}}, z_2 z_3, \ldots, \frac{z_d^2 - 1}{\sqrt{2}}\right)^\top, \tag{16}$$

and

$$\gamma := \text{argmin}_{\gamma \in \mathbb{R}^m} \mathbb{E}\left[\{\gamma^\top t(Z) - f(\mu + CZ)\}^2\right]. \tag{17}$$

In brief, $t$ is a one-to-one transformation such that the output vector has un-correlated components: $\mathbb{E}[t(Z)t^\top(Z)] = I$. That makes possible the estimation of $\gamma$ without inverting the FIM. The explicit mapping from $\gamma$ to $\eta$ depending on $(\mu, \Sigma)$ is given by the next theorem.

**Theorem 4.1** (LSVI mapping $\phi$ for full-covariance Gaussian distributions). *Let $\eta \in \mathcal{V}$ defines a Gaussian distribution $X \sim \mathcal{N}(\mu, \Sigma)$, and let $C = \mathrm{Chol}(\Sigma)$ be the Cholesky of $\Sigma$. Then, $\beta := \phi(\eta)$ is defined recursively from bottom to top by*

$$
\beta = \begin{pmatrix} \beta^{(0)} \\ \beta^{(1)} \\ \beta^{(2)} \end{pmatrix} = \begin{pmatrix} \gamma^{(0)} - \sum_{i=1}^{d} \Gamma_{i,i} - \beta^{(1),\top}\mu - \beta^{(2),\top}\operatorname{vec}\mu\mu^{\top} \\ C^{-\top}\gamma^{(1)} - 2\mu^{\top}\beta^{(2)} \\ \operatorname{vec}\left(C^{-1}\Gamma C^{-\top}\right) \end{pmatrix}, \qquad (18)
$$

*where $\gamma = \mathbb{E}[t(Z)f(\mu + CZ)]$ has subcomponents $\gamma = (\gamma^{(0)}, \gamma^{(1),\top}, \gamma^{(2),\top})^{\top}$, $\gamma^{(0)} \in \mathbb{R}$, $\gamma^{(1)} \in \mathbb{R}^d, \gamma^{(2)} \in \mathbb{R}^{d(d+1)/2}$, and where $\Gamma$ is the symmetric matrix given componentwise by $\Gamma_{i,i} = \gamma^{(2)}_{1+1/2(2d+2-i)(i-1)}/\sqrt{2}$, $\Gamma_{i,i+k} = \gamma^{(2)}_{1+1/2(2d+2-i)(i-1)+k}/2$ for $1 \le i \le d$ and $1 \le k \le d-i$. In addition, if $f$ has second-order derivatives such that $\|\mathbb{E}_X[\nabla f]\| < \infty$ and $0 \prec -\mathbb{E}_X\left[\nabla^2 f\right]$, then $\phi(\eta)$ defines a Gaussian distribution with mean and covariance given by*

$$
(\mu', \Sigma') = \left(\mu - \mathbb{E}\left[\nabla^2 f(X)\right]^{-1}\mathbb{E}\left[\nabla f(X)\right], -\mathbb{E}\left[\nabla^2 f(X)\right]^{-1}\right), \quad X \sim N(\mu, \Sigma). \qquad (19)
$$

Theorem 4.1 gives the regressor with respect to $s(X)$ of $f(X)$, as a function of $(\mu, C)$ and $\gamma$. Furthermore, all the involved operations have cost dominated by the computation of $C$, which is the same as computing products of $d \times d$ matrices, $O(d^3)$.

**Mean-field Gaussian family**  The family of mean-field Gaussian distributions is treated similarly to the previous one by removing the cross-terms $z_i z_j$ in the sufficient statistic. The total cost of the OLS computation is $\mathcal{O}(d)$. See Appendix D.3 for the explicit regression procedure.

### 4.1 Stochastic schemes tailored to Gaussian distributions

We now take advantage of the reparametrisation tricks previously introduced to derive tailored implementations of LSVI for Gaussian variational families, with optimal one-iteration cost in $d$.

An unbiased estimate of the OLS (17) is given by

$$
\hat{\gamma} = N^{-1}\sum_{i=1}^{N} t(Z_i)f(\mu + CZ_i), \qquad Z_1, \ldots, Z_N \overset{\text{i.i.d.}}{\sim} N(0, I_d). \qquad (20)
$$

We define $\hat{\eta}$ as the estimate obtained by plugging $\hat{\gamma}$ into (18) of Theorem 4.1. The mean-field case is treated in a similar manner. See Algorithms 2 and 3.

---

**Algorithm 2** LSVI-MF (mean-field Gaussian family)

---

**Require:** $(\mu_0, \sigma_0^2) : \sigma_{0,i} > 0, i \in [1, d], N \ge 1$
  $(\hat{\mu}_0, \hat{\sigma}_0^2) \leftarrow (\mu_0, \sigma_0^2)$
  $\hat{\eta}_0 \leftarrow (-\infty, -\mu/\hat{\sigma}_0^2, -\frac{1}{2\hat{\sigma}_0^2})$
  **while** not converged **do**
    $Z_1, \ldots, Z_N \sim \mathcal{N}(0, I)$
    $\hat{\gamma}_{t+1} \leftarrow \frac{1}{N}\sum_{i=1}^{N} t(Z_i)f(\hat{\mu}_t + \hat{\sigma}_t \otimes Z_i)$
    Compute $\hat{\eta}'_{t+1}$ using (43)
    $\varepsilon_t \leftarrow \texttt{stepsize}(\hat{\gamma}_{t+1}, \hat{\eta}'_{t+1}, \hat{\eta}_t, Z_{1:N})$
    $\hat{\eta}_{t+1} \leftarrow \varepsilon_t\hat{\eta}'_{t+1} + (1-\varepsilon_t)\hat{\eta}_t$
    $\hat{\mu}_{t+1} \leftarrow -\frac{1}{2}\hat{\eta}_{t+1}^{(2),-1}\hat{\eta}_{t+1}^{(1)}$
    $\hat{\sigma}_{t+1}^2 \leftarrow -\frac{1}{2}\hat{\eta}_{t+1}^{(2),-1}$
  **end while**

---

**Algorithm 3** LSVI-FC (full-covariance Gaussian family)

---

**Require:** $\mu_0, \Sigma_0 \succ 0, N \ge 1$
  $(\hat{\mu}_0, \hat{\Sigma}_0) \leftarrow (\mu_0, \Sigma_0)$
  $\hat{\eta}_0 \leftarrow (-\infty, -\Sigma^{-1}\mu, -\frac{1}{2}\operatorname{vec}\hat{\Sigma}^{-1})$
  **while** not converged **do**
    $\hat{C}_t \leftarrow \mathrm{Cholesky}(\hat{\Sigma}_t)$
    $Z_1, \ldots, Z_N \sim \mathcal{N}(0, I)$
    $\hat{\gamma}_{t+1} \leftarrow \frac{1}{N}\sum_{i=1}^{N} t(Z_i)f(\hat{\mu}_t + \hat{C}_tZ_i)$
    Compute $\hat{\eta}'_{t+1}$ using (18)
    $\varepsilon_t \leftarrow \texttt{stepsize}(\hat{\gamma}_{t+1}, \hat{\eta}'_{t+1}, \hat{\eta}_t, Z_{1:N})$
    $\hat{\eta}_{t+1} \leftarrow \varepsilon_t\hat{\eta}'_{t+1} + (1-\varepsilon_t)\hat{\eta}_t$
    $\hat{\mu}_{t+1} \leftarrow -\frac{1}{2}\hat{\eta}_{t+1}^{(2),-1}\hat{\eta}_{t+1}^{(1)}$
    $\hat{\Sigma}_{t+1} \leftarrow -\frac{1}{2}\operatorname{unvec}(\hat{\eta}_{t+1})^{(2),-1}$
  **end while**

---

# 5 Numerical experiments

We consider three examples: one where SGD may be used to minimise the KL objective, and two where it may not, because the reparameterisation trick is not possible: distributions $q$ in $\mathcal{Q}$ are discrete, $\log \pi$ is not differentiable, or because the log-derivative trick yields noisy estimates [15].

In the first example (logistic regression), we compare all three LSVI[1] instances with other gradient-based KL minimisation procedures, including ADVI, NGD, and a gradient-free procedure for Gaussian mixtures. In the second and third examples (variable selection and Bayesian synthetic likelihood, BSL), since SGD is not available, we assess the approximation error of LSVI relative to the true posterior using exact Bayesian inference.

## 5.1 Logistic regression

Given data $(x_i, y_i) \in \mathbb{R}^d \times \{-1, 1\}$, $i = 1, \ldots, n$, the posterior distribution of a logistic regression model is: $\pi(\beta) \propto p(\beta) \prod_{i=1}^n F(y_i x_i^\top \beta)$ where $F(x) = 1/(1 + e^{-x})$ and $p(\beta)$ is a (typically Gaussian) prior over the parameter $\beta$. This type of posterior is often close to a Gaussian, and is a popular benchmark in Bayesian computation [43]. See Appendix C.2 for a summary of the considered datasets and the priors.

Whenever applicable, we compare LSVI (Algorithms 1, 2, 3) with NGD and ADVI. For NGD, the gradients are obtained via JAX autodifferentiation [44] and the FIM is estimated via Monte Carlo. For ADVI, we use the standard implementations given by pyMC3 [6] and Blackjax [7] with default step size schedules (that is, a modification of Adam and RMSProp for pyMC, and comparable fixed step sizes for Blackjax). In addition, we provide a comparison of LSVI (Algorithm 1) in low dimension with the gradient-free iteration for Gaussian mixtures (GMMVI, [45]) which is a fair comparison since GMMVI and LSVI Algorithm 1 have the same complexity in this case. In addition, we illustrate the compatibility of our proposed methods with subsampling procedures for large datasets [46, 47] to reduce the cost of the log-likelihood evaluations.

Figure 1 summarises this comparison for the Pima dataset (full-covariance case). One sees that LSVI (Algorithm 1) converges essentially in one step, LSVI-FC (Algorithm 3) converges in less than 100 steps for linearly decreasing step sizes. For such a low-dimensional dataset ($d = 9$), LSVI remains competitive with LSVI-FC since it converges faster, and the matrices it needs to invert are small. LSVI performs comparably to NGD and GMMVI, but is less noisy (with or without an adaptive schedule 4). We consider larger and more challenging datasets as recommended by [43]. Figure 2 (left) does the same comparison for the MNIST dataset (mean-field covariance), In Appendix C, Figure 4 for the Sonar dataset (full-covariance) and Figure 7 for the Census-Income dataset (mean-field covariance with subsampling). This time, inverting the FIM is too costly (e.g., $2015 \times 2015$ for Sonar), so we only use the tailored schemes LSVI-MF and LSVI-FC. Section C.2 contains extra details and results for all datasets in Table 2, including runtimes and memory usage (Table 1), average cost time per iteration with respect to $N$ (Figures 4 and 5), loss vs elapsed time and classification performance (Figure 6), details on the considered schedules for the step sizes (Table 3).

## 5.2 Variable selection

Given a dataset $\mathcal{D} = (x_i, y_i)_{i=1,\ldots,n}$, $x_i \in \mathbb{R}^d$, $y_i \in \mathbb{R}$, the variable selection task in Bayesian linear regression may be modelled as $y_i = x_i^\top \operatorname{diag}(\gamma)\beta + \sigma \varepsilon_i, \varepsilon_i \sim N(0, 1)$, where $\gamma \in \{0, 1\}^d$ is a vector of inclusion variables, which is assigned a prior distribution that is a product of Bernoulli($p$); e.g., $p = 1/2$. If $(\beta, \sigma^2)$ is assigned a conjugate prior, the marginal posterior distribution $\pi(\gamma|\mathcal{D})$ (with $\beta$, $\sigma^2$ integrated out) admits a closed-form expression, the support of which is $\{0, 1\}^d$. It is therefore natural to set $\mathcal{Q}$ to the family of Bernoulli products, i.e. $q(\gamma) = \prod_{i=1}^d q_i^{\gamma_i}(1 - q_i)^{1-\gamma_i}$ with $q_i \in [0, 1]$ for $i = 1, \ldots, d$. This family is discrete, which precludes a reparametrisation trick, and the application of ADVI.

Figure 2 (right) compares the posterior inclusion probabilities, i.e. $\pi(\gamma_i = 1|\mathcal{D})$ approximated either through LSVI (Algorithm 1), or the Sequential Monte Carlo (SMC) sampler of Schäfer and Chopin [48], for the concrete dataset ($d = 92$). This dataset is challenging as it generates strong posterior correlations between the $\gamma_i$. Despite this, LSVI gives a reasonable approximation of the

---

[1]Python package: `https://github.com/ylefay/LSVI`

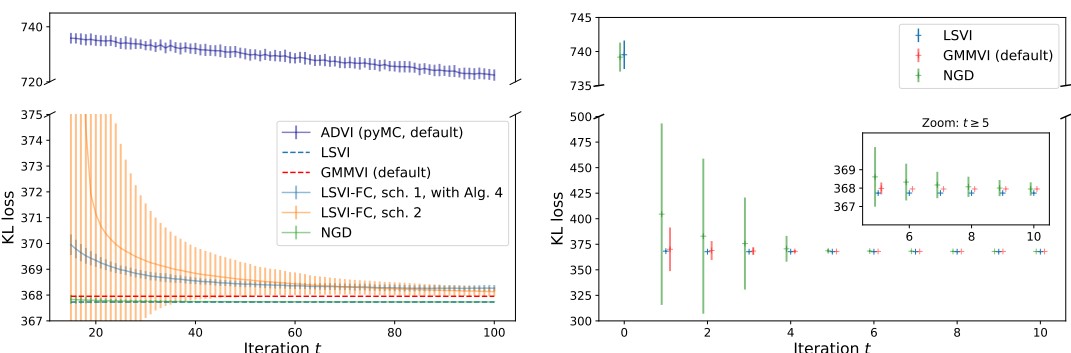

Figure 1: Logistic regression, Pima data, full-covariance approximation. KL divergence (up to an unknown constant) between the variational approximation and the posterior, as a function of the number of iterations. Left: truncated from iteration $t \geq 20$ for better readability. Right: focus on GMMVI, LSVI and NGD. Mean over 100 repetitions and one standard deviation interval (`jax.numpy.std`).

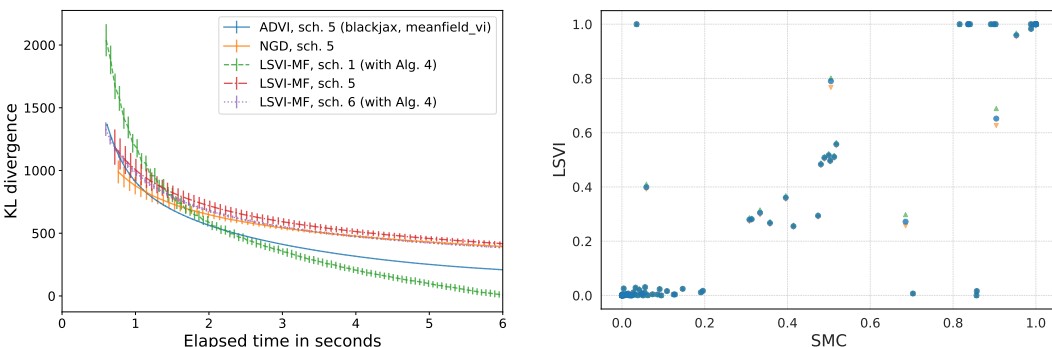

Figure 2: Left: Logistic regression, MNIST, mean-field approximation. KL divergence between the variational approximation and the posterior, as a function of the elapsed time. Truncated from iteration $t \geq 10$. Mean over 100 repetitions and one standard deviation interval. Right: Variable selection example, posterior marginal probabilities $\pi(\gamma_i = 1 | \mathcal{D})$: LSVI vs SMC. (LSVI: 100 repetitions, the min-max intervals are reported with arrows, SMC: 3 repetitions).

true posterior. To the best of our knowledge, this is the first time variational inference is implemented for variable selection using the Bernoulli product family. See Section C.3 for extra numerical results and more details on the prior, the data, and the implementation.

### 5.3   Bayesian synthetic likelihood

BSL is a popular way to perform likelihood-free inference, that is, inference on a parametric model which is described only through a simulator: one is able to sample $Y \sim P_\theta$, but not to compute the likelihood $p(y|\theta)$; see Frazier et al. [49] for a review.

BSL requires to specify $s(y)$, a low-dimensional summary of the data and assumes that $s(y) \sim N(b(\theta), \Sigma(\theta))$, leading to posterior density $\pi(\theta) \propto p(\theta) N(s(y); b(\theta), \Sigma(\theta))$, where $p(\theta)$ is the prior. Since functions $b$ and $\Sigma$ are unknown, they are replaced by empirical moments $\hat{b}(\theta)$, $\hat{\Sigma}(\theta)$, computed from simulated data. This makes BSL, and in particular its Markov Chain Monte Carlo (MCMC) implementations, particularly CPU-intensive, as the data simulator must be run many times. Furthermore, each evaluation of $\pi$ is corrupted with noise, making it impossible to differentiate $\log \pi$. Note that, in general, the data simulator is too complex to implement some form of reparametrisation trick, or the application of automatic differentiation procedures.

We consider the toad's displacement example from [50], which has been considered in various BSL papers [49, 51]. The model is parameterised by $\theta = (\alpha, \gamma, p_0) \in \mathbb{R}^+ \times \mathbb{R}^+ \times [0, 1]$. See Section C.4 for more details on the model. We implement both LSVI-MF and LSVI-FC. For the former, we use a family of truncated Gaussian distributions, while for the latter, we re-parametrise the model in terms of $\xi = f(\theta)$, where $f$ is one-to-one transform between $\Theta$ and $\mathbb{R}^d$. The top panel of Figure 3 shows that both LSVI algorithms converge quickly. The bottom panel shows that the full-covariance LSVI approximation matches the posterior obtained via MCMC, at a fraction of the CPU cost, see Table 1. Again, we refer to Section C.4 for more details on the implementation of either LSVI or MCMC.

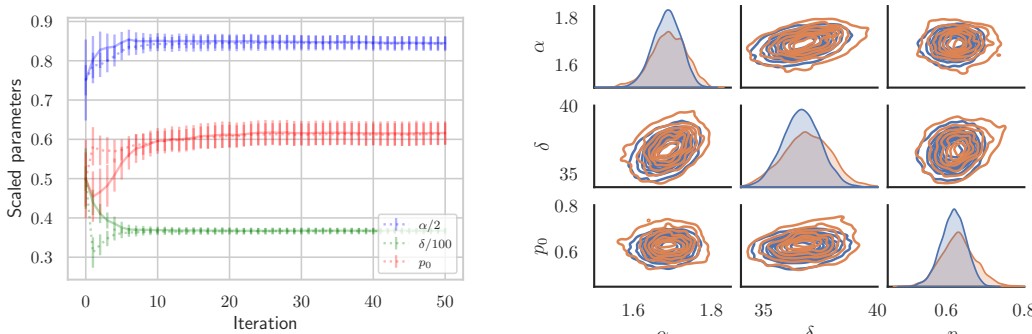

Figure 3: Left: Variational approximations of each coordinate of $\theta$ with one standard deviation interval, normalised. Truncated Gaussian: solid line. Full covariance Gaussian: dashed line. Right: Full-covariance Gaussian variational approximation (blue), MCMC approximation (orange).

## 6  Limitations

The current approach is limited to exponential families; mixture of exponential families may be tackled by adapting the expectation-maximisation approach of Arenz et al. [45], or by building on existing applications of NGD VI methods to mixtures of exponential families [19, 22]. For Gaussian approximations, if the posterior contains directions that are strongly non-Gaussian, then conditional-Gaussian strategies like integrated nested Laplace approximations may be applied [52]. In discrete exponential families, independence can be lifted by considering tree-structured dependencies, which are quite flexible, see, e.g., Wainwright and Jordan [21].

## Acknowledgments and Disclosure of Funding

The first author gratefully acknowledges partial support from the Magnus Ehrnrooth foundation. The authors thank Sam Power, Mohammad Emtiyaz Khan and anonymous reviewers for insightful remarks on a preliminary version.

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

# Contents

# A  Notations

For any vector $u \in \mathbb{R}^p$, we denote by $u^{-1} \in \mathbb{R}^p$ the componentwise inverse of $u$, and $u \otimes u'$ the componentwise product of $u$ with $u'$. For any matrix $U \in \mathbb{R}^{p \times q}$, we denote by $\mathrm{vec}(U)$ the $p \times q$ vector obtained by vertically stacking the columns of $U$, and by $\mathrm{unvec}$ the inverse operation satisfying $\mathrm{unvec}(\mathrm{vec}(U)) = U$, while $U \otimes U'$ denotes the Kronecker product between matrices. For any square matrix $U \in \mathbb{R}^{p \times p}$, let $\mathrm{diag}(U)$ be the $p$ vector composed of the diagonal components of $U$ and let $\|U\|$ be the spectral norm of $U$.

For any set $A$, $\mathcal{U}(A)$ denotes the uniform distribution over $A$. $N(\mu, \Sigma)$ denotes the Gaussian distribution with mean $\mu$ and covariance matrix $\Sigma$, and $N(\mu, \sigma^2)$ with $\sigma^2 = (\sigma_1^2, \ldots, \sigma_d^2)^\top$ denotes the Gaussian distribution with mean $\mu$ and diagonal covariance matrix $\mathrm{diag}(\sigma^2)$.

The $O$ is the usual big-$O$ notation, i.e., $A_n = O(B_n)$ for some sequences $A_n$, $B_n$, let it be reals, vectors or matrices, if there exists a constant $C > 0$ such that for $N$ large enough and all $n \geq N$, $\|A_n\| \leq C\|B_n\|$. If $\|A_n\| \geq C\|B_n\|$, we write $A_n = \Omega(B_n)$. We write $A_n = \mathcal{O}_P(1)$ for a sequence of random variables $(A_n)$ such that, for any $\varepsilon > 0$, there exists a constant $B > 0$ such that $P(\|A_n\| > B) \leq \varepsilon$ for $n$ large enough.

For any definite positive matrix $\Sigma$, we denote by $C = \mathrm{Chol}(\Sigma)$ the unique lower triangular matrix such that $CC^\top = \Sigma$.

# B  Adaptive schedule algorithm

---

**Algorithm 4** Variance control and backtracking strategy

---

**Require:** $\varepsilon' > 0$, $\eta \in \mathcal{V}$, $\eta' \in \mathbb{R}^m$, $N \geq 1$, $X_1, \ldots, X_N \overset{\text{i.i.d.}}{\sim} q_\eta$, $u > 0$
 1: $\varepsilon \leftarrow \varepsilon'$
 2: **while** $\varepsilon\eta' + (1 - \varepsilon)\eta \notin \mathcal{V}$ **do**
 3:     $\varepsilon \leftarrow \varepsilon/2$
 4: **end while**
 5: $\eta \leftarrow \varepsilon\eta' + (1 - \varepsilon)\eta$
 6: $\hat{m} \leftarrow N^{-1} \sum_{i=1}^{N} f(X_i) - \eta^\top s(X_i)$
 7: $\hat{v}^2 \leftarrow N^{-1} \sum_{i=1}^{N} (f(X_i) - \hat{m})^2$
 8: **if** $\hat{v} \geq u$ **then**
 9:     $\varepsilon \leftarrow \min(\varepsilon, u/\hat{v})$
10: **end if**
11: **return** $\varepsilon$

---

# C  Extra details on numerical experiments

## C.1  Runtime analysis

All the experiments were conducted using Python 3.13, jax 0.5 with GPU support, Cuda 12.5, and using float64. The hardware specifications are CPU AMD EPYC 7702 64-Core Processor and GPU NVIDIA A100-PCIE-40GB, except for SONAR, Census and MNIST datasets where EPYC 7713 and NVIDIA A100-PCIE-80GB were used. See Table 1.

Table 1: For all conducted experiments, runtimes and max memory usage, across 5 repetitions. $T$ is the number of iterations and $N$ the number of samples whenever applicable. LR = Logistic regression, BSL = Bayesian Synthetic likelihood, MF Gaussian = mean-field Gaussian, Gaussian = full-covariance Gaussian.

| Experiment | Runtime (seconds) | | | max resident set size (memory usage) |
|---|---|---|---|---|
| | mean (std) | min | max | (gigabytes) |
| BSL Gaussian, Alg. 1, $(N, T) = (100, 50)$ (JAX) | 72.9 ($\pm$2.8) | 71.5 | 77.8 | 1.07 |
| BSL Truncated MF Gaussian, Alg. 1, (100, 50) (JAX) | 137.5 ($\pm$0.6) | 137.3 | 138.7 | 1.05 |
| BSL MCMC, Blackjax (JAX) | 268.1 ($\pm$3.4) | 266.5 | 274.3 | 1.16 |
| Variable Selection, Alg. 1 sch. 3, $(5 \times 10^4, 25)$ | 60.8 ($\pm$0.3) | 60.3 | 61.1 | 0.42 |
| Variable Selection, SMC | 290.7 ($\pm$1.7) | 284.1 | 298 | 0.45 |
| LR Gaussian, *PIMA*, Alg. 1 sch. 3, $(10^4, 10)$ (JAX) | 1.6 ($\pm$1.4) | 1.0 | 4.1 | 1.28 |
| // NGD, $(10^4, 10)$ (JAX) | 2.2 ($\pm$1.5) | 1.51 | 4.9 | 3.29 |
| // Alg. 3 sch. 1, $(10^5, 100)$, (JAX) | 4.3 ($\pm$1.7) | 3.5 | 7.3 | 1.28 |
| // Alg. 3 sch. 2, $(10^5, 100)$, (JAX) | 3.4 ($\pm$0.1) | 3.3 | 3.6 | 1.28 |
| // PyMC ADVI, $(T = 10^4)$ | 5.9 ($\pm$0.3) | 5.5 | 6.4 | 0.84 |
| // GMMVI, $(10^4, 10)$ (TensorFlow) | 4.8 ($\pm$3.8) | 3.0 | 11.6 | 4.57 |
| LR Gaussian, *SONAR*, Alg. 3 sch. 1, $(10^5, 100)$, (JAX) | 5.1 ($\pm$0.3) | 5.0 | 5.6 | 3.11 |
| // Alg. 3 sch. 2, $(10^5, 100)$, (JAX) | 5.1 ($\pm$1.3) | 4.4 | 7.4 | 3.11 |
| // PyMC ADVI, $(10^4)$ | 9.6 ($\pm$1.3) | 7.4 | 10.5 | 3.21 |
| LR MF Gaussian, *MNIST*, Alg. 2 sch. 1, $(10^4, 500)$, (JAX) | 19.1 ($\pm$1.0) | 18.7 | 21.8 | 2.36 |
| // Alg. 2 sch. 5, $(10^4, 500)$, (JAX) | 10.4 ($\pm$0.1) | 10.3 | 10.6 | 2.36 |
| // Alg. 2 sch. 6, $(10^4, 500)$, (JAX) | 18.9 ($\pm$1.1) | 22.1 | 18.5 | 2.36 |
| // Blackjax ADVI, $(10^4, 500)$, (JAX) | 19.5 ($\pm$1.0) | 18.9 | 22.2 | 3.34 |
| // NGD (MF), sch. 5, $(10^4, 500)$ (JAX) | 25.3 ($\pm$2.2) | 29.2 | 24.1 | 4.38 |
| LR MF Gaussian, subsampling, *Census*, Alg 2, sch. 5, $(10^4, 10^4, P = 10^3)$ (JAX) | 12.9 ($\pm$0.2) | 12.8 | 13.3 | 3.30 |
| // Alg 2, sch. 6, $(10^4, 10^4, 10^3)$ (JAX) | 13.0 ($\pm$0.03) | 12.9 | 13.0 | 3.30 |
| // NGD (MF), sch. 5, $(10^4, 10^4, 10^3)$ (JAX) | 70.0 ($\pm$1.7) | 69.0 | 73.1 | 3.33 |

## C.2  Logistic regression

**Data**  The Sonar (CC BY 4.0 License) and the Census Income (CC BY 4.0 License) datasets are available in the UCI repository while the Pima dataset (CC0: Public Domain License) is in the example datasets of Python package particles (License MIT v0.4, [53, Ch. 1]) and MNIST (CC BY-SA 3.0 License) is available at `https://github.com/pjreddie/mnist-csv-png`. We use the following standard [e.g., 43] pre-processing strategy for Pima, Sonar and Census-Income datasets: we add an intercept, and we rescale the covariates so that non-binary predictors are centred with standard deviation $0.5$, and the binary predictors are centred $0$ and range $1$. For the third dataset

(MNIST dataset), we restrict ourselves to the binary classification problem by selecting pictures labelled $0$ or $8$. The gray-scale features which range between $0$ and $255$ are normalised to be between $0$ and $1$. No intercept is added. For the Census Income dataset, the categorical variables are mapped using one-hot encoding.

Table 2: Logistic regression example: summary of datasets and approximation families, in parentheses the batch-size

| Dataset | Gaussian family | $d$ | $n$ |
|---|---|---|---|
| Pima | full-covariance | 9 | 768 |
| Sonar | full-covariance | 62 | 128 |
| Census (subsampling) | mean-field | 48 | 49 000 (1000) |
| MNIST | mean-field | 784 | 11,774 |

**Prior**    For all datasets except MNIST, the prior $\pi(\beta)$ is a zero-mean Gaussian distribution with diagonal covariance matrix, and the covariances are set to $25$ for all the other covariates, except for the intercept, for which it is set to $400$. For the MNIST dataset, the prior is a Gaussian distribution with zero-mean and covariance matrix $25I_n$.

**Initialisations, schedules and number of samples**    The initialisation distributions for all datasets except MNIST are standard normal distributions. The initialisation for the MNIST dataset is $N(0, e^{-2}I_n)$. The learning schedules $(\varepsilon_t)$ are obtained via Algorithm 4 with specific inputs $(u^2, \varepsilon_t)$ summarised in Table 3 along with the number of samples $N$.

Table 3: Logistic regression setup. Left: Inputs to Algorithm 4 by dataset. Right: Schedule index reference.

| Dataset | Algorithm | Schedule input $(u^2, \varepsilon_t)$ | Samples $N$ |
|---|---|---|---|
| Pima | Alg. 1 | $(\infty, 1)$ | $10^4$ |
| Pima | NGD | $(\infty, 1/(t+1))$ | $10^4$ |
| Pima | Alg. 3 | $(10, 1), (\infty, 1/(t+1))$ | $10^5$ |
| Sonar | Alg. 3 | $(10, 1), (\infty, 1/(t+1))$ | $10^5$ |
| MNIST | Alg. 2 | $(10, 1), (\infty, 10^{-3}), (10, 10^{-3})$ | $10^4$ |
| MNIST | Blackjax (meanfield_vi), NGD (MF) | $(\infty, 10^{-3})$ | $10^4$ |
| Census | Alg. 2 | $(10, 10^{-3}), (\infty, 10^{-3})$ | $10^4$ |
| Census | NGD (MF) | $(\infty, 10^{-3})$ | $10^4$ |

| # | Schedule input $(u^2, \varepsilon_t)$ |
|---|---|
| 1 | $(10, 1)$ |
| 2 | $(\infty, 1/(t+1))$ |
| 3 | $(\infty, 1)$ |
| 4 | $(1, 1)$ |
| 5 | $(\infty, 10^{-3})$ |
| 6 | $(10, 10^{-3})$ |

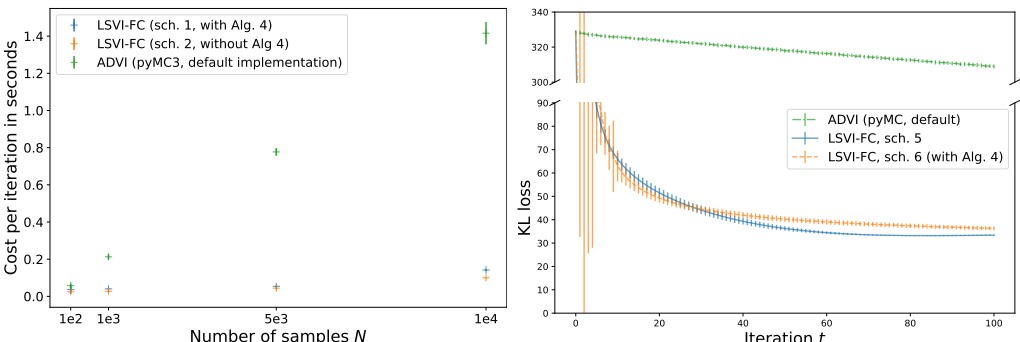

Figure 4: Logistic regression posterior, Sonar data, full-covariance approximation, LSVI-FC and ADVI implementations. Left: average cost per iteration in seconds as a function of the number of samples $N$, mean over $5$ repetitions with $2$ std interval. Right: KL divergence (up to an unknown constant) between current Gaussian variational approximation and the posterior, as a function of $t$, mean over $100$ repetitions with one standard deviation interval.

**MNIST**    The PyMC3 (License Apache 2.0 v. 5.22, [6]) implementation fails in this context, and we resort to the stochastic gradient descent (SGD) implementation in Blackjax (License Apache 2.0 v1.2.5, [7]) of the mean-field ADVI Algorithm. For SGD, we set the learning rate to $0.001$ and the

number of samples for the Monte Carlo gradient estimates to $10^4$. See Figure 5 for the average cost per iteration in seconds, and the same plot as Figure 2 with respect to elapsed time.

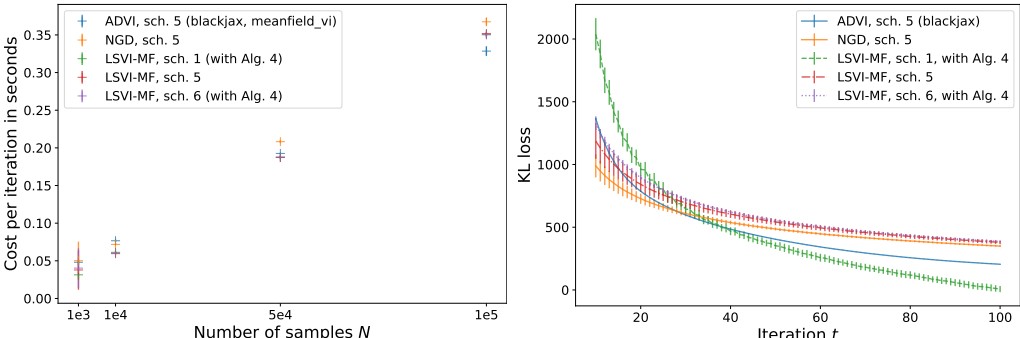

Figure 5: Logistic regression posterior, MNIST data, diagonal covariance approximation, LSVI-MF, NGD (JAX) and Blackjax (meanfield_vi) implementations. Left: average cost per iteration in seconds as a function of the number of samples $N$, mean over 5 repetitions with 2 std interval. Right: KL divergence (up to an unknown constant) between current Gaussian variational approximation and the posterior, as a function of $t$, mean over 100 repetitions with one standard deviation interval.

In addition, we provide missclassification rate for the logistic regression model using the mean (of the Gaussian approximation) as the regression parameter, see Figure 6.

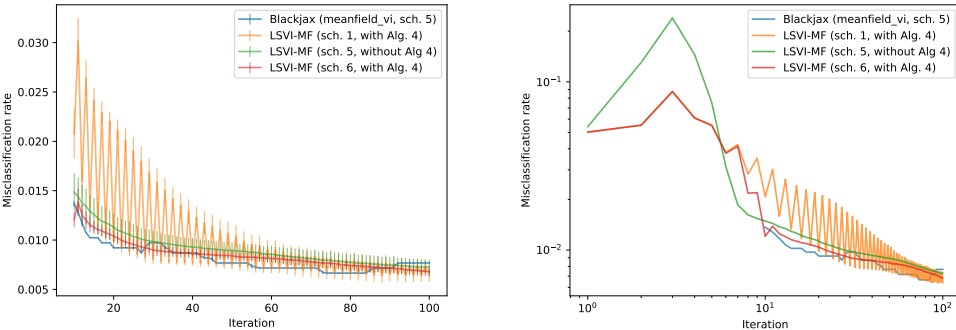

Figure 6: Logistic regression posterior, MNIST data, diagonal covariance approximation, LSVI-MF and Blackjax (meanfield_vi) implementations. Top: Misclassification rate as a function of the iterations, mean over 100 repetitions with 1 standard deviation. Bottom: same in log-log axis.

**Subsampling (Census dataset)**  At each iteration $t$, a new batch is sampled uniformly with replacement from the dataset:

$$\hat{f}(\beta) = \log \hat{\pi}(\beta) = \log p(\beta) + \sum_{i=1}^{P} \log p(y_{U_i}|x_{U_i}, \beta) + \sum_{i=1}^{P} \log p(x_{U_i}), \tag{21}$$

where $U_1, \ldots, U_P \sim \mathcal{U}(1, \ldots, n)$. A new batch is drawn at each iteration. The batch size is $P = 10^4$. We also use $\hat{f}$ for evaluating the KL loss. See Figure 7.

## C.3  Variable selection

**Dataset**  The *Concrete Compressive Strength* dataset [54] is made of 1030 observations and 8 initial predictors denoted by C, W, CA, FA, BLAST, FASH, PLAST, and A. We enrich the dataset by adding predictors computed from the existing predictors. 5 new predictors, LG_C, LG_W, LG_CA, LG_FA, LG_A, where LG_X stands for the logarithm of the corresponding feature X. The cross-product of the predictors is also added, resulting in 78 new predictors. Finally, we add an intercept. The total of possible predictors is $d = 92$.

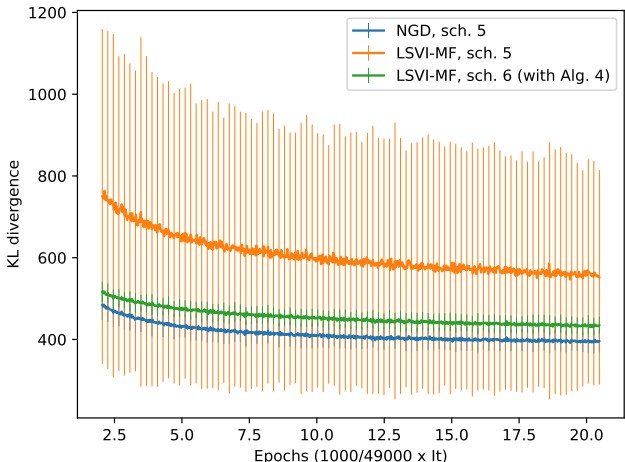

Figure 7: Logistic regression posterior. KL loss for the Census-Income dataset (mean-field, with subsampling), mean over 100 repetitions with 1 standard deviation.

**Prior**  The hierarchical prior on $\beta, \sigma^2, \gamma$ is given by

$$\pi(\beta \mid \sigma, \gamma, Z) = N\left(0, \sigma^2 v^2 \operatorname{diag}(\gamma)\right), \quad \pi(\sigma^2) = \operatorname{InvGamma}(w/2, \lambda w/2), \quad \pi(\gamma) = \mathcal{U}(\{0,1\}^d).$$

We follow the recommendations of [55] by setting the hyperparameters to $w = 4.0$, $\lambda = \hat{\sigma}_1^2$ and $v^2 = 10/\lambda$, where $\hat{\sigma}_1^2$ is the variance estimate of the residuals for the saturated linear model $\gamma = (1, \dots, 1)$.

**Close-form expression for** $\pi(\gamma|\mathcal{D})$  For a model $\gamma \in \{0,1\}^d$, let $Z_\gamma = [Z_i]_{i/\gamma_i=1}$ be the selected covariates and let $b_\gamma = Z_\gamma^\top y$. Consider the Cholesky decomposition $C_{\gamma,v} C_{\gamma,v}^\top = Z_\gamma^\top Z_\gamma + v^{-2} I_{\|\gamma\|_1}$, and define the least squares estimate for the residuals based on the model given by $\gamma$, $\hat{\sigma^2}_{\gamma,v} = \frac{1}{d}(y^\top y - (C_{\gamma,v}^{-1} b_\gamma)^\top (C_{\gamma,v}^{-1} b_\gamma))$. Then, the log-posterior for $\gamma$ up to the log-partition constant is given by

$$\log \pi(\gamma \mid \mathcal{D}) = -\sum_{i=1}^{\|\gamma\|_1} \log c_{i,i}^{(\gamma,v)} - \|\gamma\|_1 \log(v) - \frac{w+d}{2} \log(w\lambda/d + \hat{\sigma^2}_{\gamma,v}).$$

**SMC, extra numerical results**  As a benchmark, we compute the posterior marginal probabilities of inclusion using a waste-free variant of the tempering SMC algorithm of [48] with chain length $P = 10^4$ and $N = 10^5$ particles.

Given any probability vector $p \in [0,1]^d$, we plot the histogram of the variable $\log(\pi^*(\gamma)/q(\gamma \mid p))$ with $\gamma \sim q(\cdot \mid p)$ (Bernoulli product). The pendant for the SMC discrete measure is obtained by replacing $q$ with the SMC empirical measure $\hat{\pi}^*$. In Figure 8 we plot the histograms when $\gamma$ is distributed according to the SMC empirical distribution $\hat{\pi}^*$, and when $\gamma$ is distributed according to three different mean-field Bernoulli distributions $\gamma \sim q(\cdot \mid p)$: i) $p = \left(1, \frac{1}{2}, \dots, \frac{1}{2}\right)$, i.e., the intercept is always included and the other coordinates has 0.5 probability to be included, ii) the LSVI estimates, and iii) the marginal posterior probabilities estimated via SMC.

### C.4  BSL and toads displacement model

**Model**  The model assumes that $M$ toads move along a one-dimensional axis during $D$ days. For any day $1 \le t \le D$, the toad labelled by $1 \le i \le M$, has observed position $y_{i,t}$. During the night of day $t+1$, the toad moves according to an overnight displacement, $\delta y_{i,t}$ which is assumed to be a Lévy-alpha stable distribution with stability parameter $\alpha$ and scale parameter $\delta$. With probability $p_0$, the toad takes refuge at $y_{i,t} + \delta y_{i,t}$. With probability $1 - p_0$, the toad moves back to one the

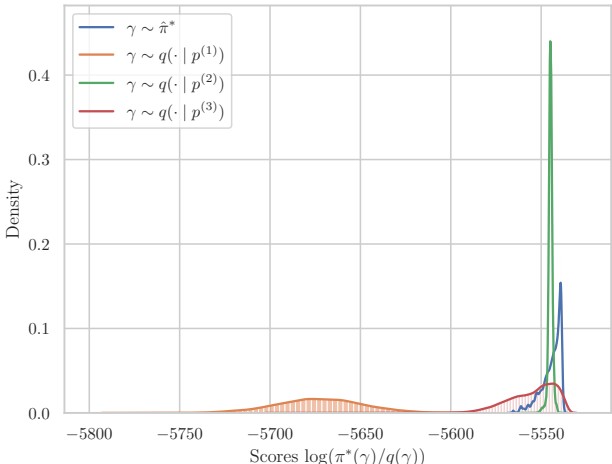

Figure 8: Variable selection example: distribution of scores $\log \pi^*(\gamma)/q(\gamma)$ when $\gamma \sim q = \hat{\pi}^*$, when $\gamma \sim q = q(\cdot \mid p^{(i)})$ with $p^{(i)}$ given either by i), ii) or iii)).

previously explored sites $y_{i,t'}$ with $t'$ chosen uniformly in $1, \ldots, t$. Finally, for any day $1 \leq t < D$ the observed position is

$$y_{i,t+1} = B_{i,t}(y_{i,t} + \delta y_{i,t}) + (1 - B_{i,t})y_{i,t'}, \tag{22}$$

with $B_{i,t} \sim \mathrm{Ber}(p_0)$, $t' \sim \mathcal{U}\{1, \ldots, t\}$ and $\delta y_{i,t} \sim$ Lévy-alpha$(\alpha, \delta)$, all variables being mutually independent. The initial position $y_i^{(1)}$ is set to $\delta y_i^{(0)} \sim$ Lévy-alpha$(\alpha, \delta)$. The model is parametrised by $\theta = (\alpha, \delta, p_0) \in [1, 2] \times [0, 100] \times [0, 0.9] := \Theta$. Simulating from the previous model yields the observed data $Y = (y_{i,t})_{1 \leq t \leq D, 1 \leq i \leq M}$.

**Summary statistic** The summary statistic is the concatenation of 4 sets of statistics of size 12 resulting in a total statistic of dimension 48. Each subset is computed from the displacement information of lag $l$ for $l \in \{1, 2, 4, 8\}$, denoted by $Y_l = (|y_{i,t} - y_{i,t+1}|)_{1 \leq t \leq D-1, 1 \leq i \leq M}$. If the displacement from $t$ to day $t + 1$ of the toad $i$, $Y_l^{(i,d)} = |y_{i,t} - y_{i,t+1}|$ is less than 10, it is assumed the toad has not moved. The first statistic is the number of pairs $(i, t)$ such that $Y_l^{(i,t)} \leq 10$. We then compute the median displacement and the log difference between adjacent $p$-quantiles with $p = 0, 0.1, \ldots, 1$ for all the displacements greater than 10.

**Truncated Gaussian distributions approximation** The dataset $Y$ is generated with $(M, D) = (66, 63)$ and underlying $\theta^* = (1.7, 35, 0.6)$. The mean and covariance estimates are obtained with $P = 100$ samples for each evaluation of the synthetic likelihood. We follow the methodology of [49] and use [56] shrinkage covariance estimate given by $\hat{\Sigma} = \hat{D}^{1/2}(\gamma \hat{C} + (1 - \gamma)I)\hat{D}^{1/2}$ where $\hat{D}$ is the estimated correlation matrices and $\gamma = 0.5$ is the regularization parameter. The prior distribution is the uniform distribution over $\Theta$. The variational family is the set of truncated Gaussian distributions over $\Theta$ with diagonal covariances. The initial distribution has mean $\mu = (1.5, 50, 0.5)$ and diagonal covariances $\sigma^2 = (0.05, 10, 0.01)$. We run Algorithm 1 with $N = 100$ samples and $T = 50$ iterations, the step sizes are obtained by Algorithm 4 with $u = 1$ and linearly decreasing step sizes.

**Full-covariance Gaussian distributions on transformed parameters** To constrain the parameters $\theta$, we perform inference on the transformed parameters $g(\theta) = \mathrm{logit}(g_i(\theta_i))$ with $g_i(\theta_i) = (\theta_i - a_i)/b_i$, with $a_i, b_i$ such that $g_i$ scales $\theta_i$ to $[0, 1]$. The prior distribution on the unconstrained parameters $\theta'$ is $\mathbb{1}_\Theta \circ g^{-1}(\theta') \times |\nabla g^{-1}(\theta')|$. The variational family is the set of full-covariance Gaussian distributions. The initial distribution for $\theta'$ has mean $\mu' = (0, 0, 0)$ and covariance matrix $\Sigma' = \mathrm{diag}(0.1, 0.1, 0.1)$. The benchmark is obtained via MCMC with random walk step

$N(0, 0.1I_3)$, the acceptance rate over the chain of length $10^4$ is roughly 31%, excluding the first $10^3$ states.

## D Proofs

### D.1 First order condition and critical points of the uKL objective

*Proof of Proposition 2.3.* Injecting $\pi = \exp(f)$ and $q_\eta = \exp(\eta^\top s)$ into the objective function, we obtain

$$\text{uKL}(q_\eta \mid \pi) = \int (\eta^\top s - f)q_\eta + \int \pi - \int \exp(\eta^\top s). \tag{23}$$

Using (23), we have

$$\nabla_\eta \text{uKL} = \int ss^\top q_\eta \eta - \int sfq_\eta. \tag{24}$$

Writing the first-order optimality condition for the following minimisation problem

$$\eta^* \in \text{argmin}_{\eta \in \mathcal{V}} \text{uKL}(q_\eta \mid \pi) \tag{25}$$

and applying (24), then dividing by $Z_\eta < \infty$, yield $\mathbb{E}_\eta[ss^\top]\eta = \mathbb{E}_\eta[fs]$. Let $s = (1, \bar{s}^\top)^\top$ be some fixed statistic with first component 1. Assume that $\eta = (\eta^{(0)}, \bar{\eta}^\top)^\top \in \mathcal{V}$ is a critical point, i.e., $\nabla_\eta \text{uKL}(q_\eta \mid \pi) = 0$. We have

$$\partial_{\eta^{(0)}} \text{uKL}(q_\eta \mid \pi) = \eta^\top \int sq_\eta - \int fq_\eta. \tag{26}$$

Injecting $\eta^\top s = \eta^{(0)} + \bar{\eta}^\top \bar{s}$ into (26), setting $\nabla_{\eta^{(0)}} \text{uKL}(q_\eta \mid \pi) = 0$ and normalising by $Z_\eta$ yields

$$\eta^{(0)} = \mathbb{E}_\eta \left[ f - \bar{\eta}^\top \bar{s} \right], \tag{27}$$

from the definition of the KL divergence, we deduce

$$\eta^{(0)} = -\text{KL}(\bar{q}_{\bar{\eta}} \mid \bar{\pi}) + \log \left( Z(\pi) / \int_{\mathcal{X}} \exp(\bar{\eta}^\top \bar{s}) \right). \tag{28}$$

$\square$

*Proof of Proposition 2.2.* We have

$$\text{KL}(\bar{q}_{\bar{\eta}} \mid \bar{\pi}) = Z_{\bar{\eta}}^{-1} \int e^{\bar{\eta}^\top \bar{s}}(\bar{\eta}^\top \bar{s} - f) - \log Z_{\bar{\eta}} + \log Z_\pi.$$

Computing the gradient of the KL requires computing the gradients of $Z_\eta$, $\log Z_\eta$, and $Z_\eta^{-1}$. We have $\nabla_{\bar{\eta}} Z_{\bar{\eta}} = \int \bar{s} e^{\bar{\eta}^\top \bar{s}}$, and $\nabla_{\bar{\eta}} \log(Z_{\bar{\eta}}) = Z_{\bar{\eta}}^{-1} \nabla_{\bar{\eta}} Z_{\bar{\eta}} = \mathbb{E}_\eta[\bar{s}]$. Similarly, $\nabla_{\bar{\eta}} Z_{\bar{\eta}}^{-1} = -Z_{\bar{\eta}}^{-2} \nabla_{\bar{\eta}} Z_{\bar{\eta}} = -Z_{\bar{\eta}}^{-1} \mathbb{E}_\eta[\bar{s}]$. Then, using the previous equalities, the gradient of the KL is

$$\nabla_{\bar{\eta}} \text{KL}(\bar{q}_{\bar{\eta}} \mid \bar{\pi}) = \nabla_{\bar{\eta}} \left( Z_{\bar{\eta}}^{-1} \int e^{\bar{\eta}^\top \bar{s}}(\bar{\eta}^\top \bar{s} - f) \right) - \nabla_{\bar{\eta}} \log(Z_{\bar{\eta}})$$

$$= \nabla_{\bar{\eta}} Z_{\bar{\eta}}^{-1} \times \int e^{\bar{\eta}^\top \bar{s}}(\bar{\eta}^\top \bar{s} - f) + Z_{\bar{\eta}}^{-1} \nabla_{\bar{\eta}} \left( \int e^{\bar{\eta}^\top \bar{s}}(\bar{\eta}^\top \bar{s} - f) \right) - \nabla_{\bar{\eta}} \log(Z_{\bar{\eta}})$$

$$= -\mathbb{E}_\eta[\bar{s}]\mathbb{E}_\eta[\bar{\eta}^\top \bar{s} - f] + \mathbb{E}_\eta[\bar{s}\bar{s}^\top \bar{\eta} - \bar{s}f + s] - \mathbb{E}_\eta[s]$$

$$= -\mathbb{E}_\eta[\bar{s}]\mathbb{E}_\eta \left[ \bar{\eta}^\top \bar{s} - f \right] + \mathbb{E}_\eta \left[ \bar{s}\bar{s}^\top \bar{\eta} - \bar{s}f \right]. \tag{29}$$

Now, let us compute the gradient of the uKL objective with respect to $\eta = (\eta^{(0)}, \bar{\eta}^\top)^\top$. Using

$$ss^\top = \begin{pmatrix} 1 & \bar{s}^\top \\ \bar{s} & \bar{s}\bar{s}^\top \end{pmatrix} \tag{30}$$

to expand (24) yields

$$\partial_{\eta^{(0)}} \text{uKL}(q_\eta \mid \pi) = \int (\eta^{(0)} + \bar{s}^\top \bar{\eta})q_\eta - \int fq_\eta,$$

$$\nabla_{\bar{\eta}} \text{uKL}(q_\eta \mid \pi) = \int (\bar{s}\eta^{(0)} + \bar{s}\bar{s}^\top \bar{\eta})q_\eta - \int \bar{s}fq_\eta. \tag{31}$$

Assume that $\nabla_\eta \text{uKL}(q_\eta \mid \pi) = 0$, then from (31), we obtain $\mathbb{E}_\eta[\bar{s}\eta^{(0)} + \bar{s}\bar{s}^\top \bar{\eta}] - \mathbb{E}[\bar{s}f] = 0$. Reinjecting the previous inequality into the gradient of the KL (29) yields

$$\nabla_{\bar{\eta}} \text{KL}(\bar{q}_{\bar{\eta}} \mid \bar{\pi}) = -\mathbb{E}_\eta[\bar{s}]\mathbb{E}_\eta[\bar{s}^\top \bar{\eta} - f] - \mathbb{E}_\eta[\bar{s}]\eta^{(0)}. \tag{32}$$

Injecting the expression for $\eta^{(0)}$ (27) into (32) yields $\nabla_{\bar{\eta}} \text{KL}(\bar{q}_{\bar{\eta}} \mid \bar{\pi}) = 0$. Conversely, the previous computations show that if $\nabla_{\bar{\eta}} \text{KL}(\bar{q}_{\bar{\eta}} \mid \bar{\pi}) = 0$ and $\partial_{\eta^{(0)}} \text{uKL}(q_\eta \mid \pi) = 0$, then $\nabla_\eta \text{uKL}(q_\eta \mid \pi) = 0$. $\qquad\square$

## D.2 The exact LSVI is a natural gradient descent

*Proof of Proposition 2.4.* Assumption 2.1 ensures that for any $\eta \in \mathcal{V}$, $F_\eta$ is invertible (minimality assumption), and ensures the differentiability of all the involved functions (regularity). Let us denote by $\nabla_\eta l$ the Jacobian of $\eta \mapsto l(\eta)$. Let $\eta \in \mathcal{V}$, using (24), we have

$$\nabla l_\eta(\eta) = Z_\eta(F_\eta \eta - z_\eta), \tag{33}$$

where $Z_\eta = Z(q_\eta)$ is the normalisation constant of $q_\eta$. Let $(\eta_t)$ be the sequence obtained via natural gradient descent given by (6). Then, by (6) and (33), we have

$$
\begin{aligned}
\eta_{t+1} &= \eta_t - \frac{\varepsilon_t}{Z_{\eta_t}} F_{\eta_t}^{-1} \nabla_\eta l(\eta_t) \\
&= \eta_t - \varepsilon_t F_{\eta_t}^{-1}(F_{\eta_t}\eta_t - z_{\eta_t}) \\
&= (1 - \varepsilon_t)\,\eta_t + \varepsilon_t F_{\eta_t}^{-1} z_{\eta_t} \\
&= (1 - \varepsilon_t)\,\eta_t + \varepsilon_t \phi(\eta_t).
\end{aligned}
\tag{34}
$$

Thus, the LSVI iteration with learning schedule $(\varepsilon_t)$ given by (5) is the natural gradient descent $(\eta_t)$ with learning schedule $(\varepsilon_t/Z_{\eta^{(t)}})$ given by (6). Let us now prove (7). We have

$$
\begin{aligned}
\nabla_\eta \omega &= \nabla_\eta^2 Z \\
&= \int ss^\top q_\eta \, \mathrm{d}\mu \\
&= Z_\eta F_\eta.
\end{aligned}
\tag{35}
$$

By the chain rule and (35), the Jacobian of $\eta \mapsto l(\eta)$ is

$$
\begin{aligned}
\nabla_\eta l &= \nabla_\eta \omega \times \nabla_\omega l \\
&= Z_\eta F_\eta \times \nabla_\omega l.
\end{aligned}
\tag{36}
$$

Finally, injecting (36) into (6) yields

$$\eta_{t+1} = \eta_t - \varepsilon_t \nabla_\omega l(\omega(\eta_t)), \tag{37}$$

which is (7). This shows the first equivalence. Let $\omega_0 \in \mathcal{W}$, and define $(\omega_t)$ as given by (8). The first order condition on the minimisation problem (8) yields

$$\nabla Z^\star(\omega_{t+1}) = \nabla_\omega Z^\star(\omega_t) - \varepsilon_t \nabla_\omega l(\omega_t), \tag{38}$$

but $\nabla Z^*(\omega_t) = \eta(\omega_t) = \eta_t$, thus (38) is exactly (7). $\qquad\square$

## D.3 The exact LSVI mapping for mean-field Gaussian distributions

Let $s(x) \coloneqq (1, x, x^2)^\top$ where $x = (x_1, \ldots, x_d)$ and $x^2 = (x_1^2, \ldots, x_d^2)$. The set of admissible natural parameters is given by $\mathcal{V} = \mathbb{R} \times \mathbb{R}^d \times (\mathbb{R}^-\backslash\{0\})^d \times \subset \mathbb{R}^m$, $m = 2d + 1$. Let $\eta = (\eta^{(0)}, \eta^{(1),\top}, \eta^{(2),\top})^\top \in \mathcal{V}$. The natural mapping from $\eta$ to $(\mu, \sigma^2)$ is given by $T(\eta) \coloneqq \left(-\frac{1}{2}\eta^{(1)} \otimes \eta^{(2),-1}, -\frac{1}{2}\eta^{(2),-1}\right)$ where $\otimes$ is the componentwise product and $\eta^{(2),-1}$ is the componentwise inverse of $\eta^{(2)}$.

**Lemma D.1** (Reparametrisation of the regression in the mean-field case). *Let $X \sim N(\mu, \sigma^2)$ be a mean-field Gaussian distribution with $\mu, \sigma \in \mathbb{R}^d$, and $\sigma_i > 0$ for all $i \in \{1, \ldots, d\}$. Let $\eta = (\eta^{(0)}, \eta^{(1),\top}, \eta^{(2),\top}) \in \mathcal{V}$, $\eta^{(0)} \in \mathbb{R}$, $\eta^{(1)} \in \mathbb{R}^d$, $\eta^{(2)} \in \mathbb{R}^d$ be the natural parameter associated with $X$ for the statistic $s : x \in \mathbb{R}^d \mapsto (1, x, x^2)^\top \in \mathbb{R}^{1+2d}$. Let $t$ be given by (42). For any $z \in \mathbb{R}^d$, let*

$x(z) = \mu + \sigma \otimes z$, if $Z \sim N(0, I)$, then $x(Z) \sim N(\mu, \sigma^2)$. Let $\gamma = (\gamma^{(0)}, \gamma^{(1),\top}, \gamma^{(2),\top})^\top \in \mathbb{R}^{2d+1}$, $\gamma^{(0)} \in \mathbb{R}$, $\gamma^{(1)} \in \mathbb{R}^d$, $\gamma^{(2)} \in \mathbb{R}^d$ be defined componentwise by

$$\gamma = \begin{pmatrix} \eta^{(0)} + \eta^{(1),\top}\mu + \eta^{(2),\top}(\mu_j^2 + \sigma_j^2)_j \\ \eta^{(1)} \otimes \sigma + 2\eta^{(2)} \otimes \mu \otimes \sigma \\ \sqrt{2}\eta^{(2)} \otimes \sigma^2 \end{pmatrix}. \tag{39}$$

Then, for any $z \in \mathbb{R}^d$

$$\gamma^\top t(z) = \eta^\top s(x(z)). \tag{40}$$

*Proof.* Let us identify $\gamma$ such that (40) is satisfied. Suppose that for all $z \in \mathbb{R}^d$, we have (40), then

$$\begin{aligned}
\eta^\top s(x) &= \eta^{(0)} + \eta^{(1),\top}x + \eta^{(2),\top}x^2 \\
&= \eta^{(0)} + \eta^{(1),\top}\mu + \eta^{(1),\top}(\sigma \otimes z) + \eta_2^\top(\mu_j^2)_j \\
&\quad + 2\eta^{(2),\top}(\mu_j\sigma_j z_j)_j + \eta^{(2),\top}(\sigma_j^2 z_j^2)_j \\
&= \underbrace{\eta^{(0)} + \eta^{(1),\top}\mu}_{\text{terms in group 1}} + \underbrace{(\eta^{(1)} \otimes \sigma)^\top z}_{\text{term in group 2}} + \underbrace{\eta^{(2),\top}(\mu_j^2)_j}_{\text{term in group 1}} \\
&\quad + \underbrace{2(\eta^{(2)} \otimes \sigma \otimes \mu)^\top z}_{\text{term in group 2}} + \underbrace{(\eta^{(2)} \otimes \sigma^2)^\top 1_d}_{\text{term in group 1}} \\
&\quad + \underbrace{(\eta^{(2)} \otimes \sigma^2)^\top(z_j^2 - 1)_j}_{\text{term in group 3}} \\
&= \gamma^\top t(z).
\end{aligned} \tag{41}$$

By identifying the factors in front of 1 (group 1), the $z_j$'s (group 2), and the $z_j^2$'s (group 3), we obtain (39). By injecting (39) into (40), the equality is satisfied. □

**Theorem D.2** (LSVI mapping $\phi$ for the mean-field Gaussian distributions). *Let $X \sim N(\mu, \sigma^2)$, and $\eta \in \mathcal{V}$ be the corresponding natural parameter and let $t$ be given by*

$$t(z) := \left(1, z^\top, \frac{z_1^2 - 1}{\sqrt{2}}, \ldots, \frac{z_d^2 - 1}{\sqrt{2}}\right)^\top. \tag{42}$$

*Then, the LSVI mapping $\beta := \phi(\eta)$ is defined recursively bottom to top by*

$$\beta = \begin{pmatrix} \gamma^{(0)} - \beta^{(1),\top}\mu - \beta^{(2),\top}(\mu^2 + \sigma^2) \\ \gamma^{(1)} \otimes \sigma^{-1} - 2\beta^{(2)} \otimes \mu \\ \gamma^{(2)} \otimes (\sqrt{2}\sigma^2)^{-1} \end{pmatrix} \tag{43}$$

*and $\gamma := \mathbb{E}[t(Z)f(\mu + \sigma \otimes Z)]$, with subcomponents $\gamma = (\gamma^{(0)}, \gamma^{(1),\top}, \gamma^{(2),\top})^\top$, $\gamma^{(0)} \in \mathbb{R}$, $\gamma^{(1)}, \gamma^{(2)} \in \mathbb{R}^d$. In addition, if $f$ admits second-order derivatives such that $\mathbb{E}_X[f] < \infty$, $\|\mathbb{E}_X[\nabla f]\| < \infty$, and $0 \prec -\mathbb{E}_X\left[\text{Diag}(\nabla^2 f)\right]$, then $\phi(\eta)$ defines a Gaussian distribution with mean and variance given by*

$$(\mu', \sigma'^2) = \left(\mu - \left(\mathbb{E}\left[\text{diag}(\nabla^2 f)(X)\right]\right)^{-1} \otimes \mathbb{E}\left[\nabla f(X)\right], -\left(\mathbb{E}\left[\text{diag}(\nabla^2 f)(X)\right]\right)^{-1}\right). \tag{44}$$

*Proof.* We know that $\phi(\eta)$ realises the minimum of the OLS objective (4), i.e.,

$$\phi(\eta) = \text{argmin}_{\beta \in \mathbb{R}^m} \mathbb{E}_{X \sim N(\mu, \sigma^2)}\left[\left(\beta^\top s(X) - f(X)\right)^2\right]. \tag{45}$$

Using Lemma D.1, we can rewrite the regression objective with covariates given by $s$ into a regression with covariates given by $t$. Using the notations of Lemma D.1, we let $\gamma$ be given such that $\gamma^\top t(z) = \beta^\top s(x(z))$ for all $z \in \mathbb{R}^d$, and where $\beta = \phi(\eta)$ is the unique minimizer of the OLS objective (45). Then,

$$\begin{aligned}
\gamma &= \text{argmin}_{\gamma \in \mathbb{R}^m} \mathbb{E}_Z\left[\left(\gamma^\top t(Z) - f(\mu + \sigma \otimes Z)\right)^2\right] \\
&= \left(\mathbb{E}_Z\left[tt^\top(Z)\right]\right)^{-1}\mathbb{E}_Z\left[t(Z)f(\mu + \sigma \otimes Z)\right] \\
&= \mathbb{E}_Z\left[t(Z)f(\mu + \sigma \otimes Z)\right],
\end{aligned} \tag{46}$$

since $\mathbb{E}_Z \left[ t t^\top (Z) \right] = I_m$. Inverting the relation (39) given by Lemma D.1 between $\gamma$ and $\beta$, which is possible since all the $\sigma_i$'s are strictly positive, we obtain

$$
\beta = \begin{pmatrix} \gamma^{(0)} - \beta^{(1),\top} \mu - \beta^{(2),\top} (\mu^2 + \sigma^2) \\ \gamma^{(1)} \otimes \sigma^{-1} - 2\beta^{(2)} \otimes \mu \\ \gamma^{(2)} \otimes \left( \sqrt{2}\sigma^2 \right)^{-1} \end{pmatrix}. \tag{47}
$$

But $\beta = \phi(\eta)$, this proves the first statement (43) of Theorem D.2. For the second statement, assume that $f$ admits second-order derivatives. Using Stein's Lemma and (46), we obtain,

$$
\begin{aligned}
\gamma &= \mathbb{E}_{Z \sim N(0,I_d)}[t(Z)f(\mu + \sigma \otimes Z)] \\
&= \begin{pmatrix} \mathbb{E}_Z \left( f(\mu + \sigma \otimes Z) \right) \\ \sigma \otimes \mathbb{E}_Z (\nabla f)(\mu + \sigma \otimes Z) \\ \frac{1}{\sqrt{2}} \left( \sigma^2 \otimes \mathbb{E}_Z \operatorname{diag}(\nabla^2 f)(\mu + \sigma \otimes Z) \right) \end{pmatrix}.
\end{aligned} \tag{48}
$$

Injecting (48) into (43), we obtain for

$$
\phi(\eta) = \begin{pmatrix} \phi(\eta)_0 \\ \mathbb{E}_X (\nabla f)(X) - \mu \otimes \mathbb{E} \operatorname{diag}(\nabla^2 f)(X) \\ \frac{1}{2} \mathbb{E} \operatorname{diag}(\nabla^2 f)(X) \end{pmatrix}. \tag{49}
$$

Using the natural mapping $T(\eta) = \left( -\frac{1}{2}\eta^{(1)}\eta^{(2),-1}, -\frac{1}{2}\eta^{(2),-1} \right)$, we obtain (44). $\qquad \square$

## D.4 The exact LSVI mapping for Gaussian distributions

**Lemma D.3** (Reparametrisation of the regression in the full-covariance case). *Let $X \sim N(\mu, \Sigma)$ be a Gaussian distribution with $\mu \in \mathbb{R}^d$, and $\Sigma \succ 0$. Let $\eta \in \mathcal{V}$ be the natural parameter associated with $X$ for the statistic $s : x \in \mathbb{R}^d \mapsto (1, X, (\operatorname{vec} XX^\top)^\top)^\top \in \mathbb{R}^{1+d+d^2}$. Let $t$ be given by (16). For any $z \in \mathbb{R}^d$, let $x(z) = \mu + Cz$ with $C \in \mathbb{R}^{d \times d}$ such that $CC^\top = \Sigma$. If $Z \sim N(0, I_d)$, then $x(Z) \sim N(\mu, \Sigma)$, and for any $z \in \mathbb{R}^d$*

$$
\gamma^\top t(z) = \eta^\top s(x(z)), \tag{50}
$$

*with $\gamma = (\gamma^{(0)}, \gamma^{(1),\top}, \gamma^{(2),\top})^\top \in \mathbb{R}^{1+d+d(d+1)/2}$. Furthermore, the components of $\gamma$, $\gamma^{(0)} \in \mathbb{R}$, $\gamma^{(1)} \in \mathbb{R}^d$, $\gamma^{(2)} \in \mathbb{R}^{d(d+1)/2}$ are given by*

$$
\gamma = \begin{pmatrix} \eta^{(0)} + \eta^{(1),\top} \mu + \eta^{(2),\top} \operatorname{vec} \mu\mu^\top + \sum_{i=1}^n \Gamma_{i,i} \\ C^\top \eta^{(1)} + 2 \left( \mu \otimes C \right)^\top \eta^{(2)} \\ \gamma^{(2)} \end{pmatrix}, \tag{51}
$$

*where*

$$
\Gamma = \operatorname{unvec} \left( (C \otimes C)^\top \eta^{(2)} \right), \tag{52}
$$

*and*

$$
\gamma^{(2)} = \left( \sqrt{2}\Gamma_{1,1}, 2\Gamma_{1,2}, \ldots, 2\Gamma_{1,d}, \sqrt{2}\Gamma_{2,2}, 2\Gamma_{2,3}, \ldots, 2\Gamma_{2,d}, \ldots, \sqrt{2}\Gamma_{d,d} \right)^\top. \tag{53}
$$

*Proof.* The proof is similar to the proof of Lemma D.1. Let us rewrite the regression with respect to $Z$. Let $\eta = (\eta^{(0)}, \eta^{(1),\top}, \eta^{(2),\top})^\top \in \mathbb{R}^{1+d+d^2}$ with $\eta^{(0)} \in \mathbb{R}$, $\eta^{(1)} \in \mathbb{R}^d$, $\eta^{(2)} \in \mathbb{R}^{d^2}$. Let $X = \mu + CZ$ with $C$ such that $CC^\top = \Sigma$. Rewriting the linear regression on $s(X)$ with $s(Z)$, we have

$$
\begin{aligned}
\eta^\top s(X) &= \eta^{(0)} + \eta^{(1),\top} \mu + \eta^{(1),\top} CZ + \eta^{(2),\top} \operatorname{vec} \mu\mu^\top \\
&\quad + \eta^{(2),\top} \operatorname{vec} \mu Z^\top C^\top + \eta^{(2),\top} \operatorname{vec} CZ\mu^\top \\
&\quad + \eta^{(2),\top} \operatorname{vec} CZZ^\top C^\top \\
&= \hat{\gamma}^\top s(Z),
\end{aligned} \tag{54}
$$

where $\hat{\gamma} = (\hat{\gamma}^{(0)}, \hat{\gamma}^{(1),\top}, \hat{\gamma}^{(2),\top})^\top \in \mathbb{R}^{1+d+d^2}$ are left to be identified. By identifying the quadratic terms in (54), we have for $\hat{\gamma}^{(2)}$

$$
\eta^{(2),\top} \operatorname{vec} CZZ^\top C^\top = \eta^{(2),\top} (C \otimes C) \operatorname{vec} ZZ^\top = \hat{\gamma}^{(2),\top} \operatorname{vec} ZZ^\top, \tag{55}
$$

where we used $\mathrm{vec}\, ABC = (C^\top \otimes A)\,\mathrm{vec}\, B$. Thus,

$$\eta^{(2)} = (C \otimes C)^{-\top}\hat{\gamma}^{(2)} = (C^{-1} \otimes C^{-1})^\top \hat{\gamma}^{(2)}, \tag{56}$$

where we used $(A \otimes B)^{-1} = A^{-1} \otimes B^{-1}$. For $\hat{\gamma}^{(1)}$, expanding the linear term in (54), we have

$$\begin{aligned}
\eta^{(1),\top}Cz + \eta^{(2),\top}\,\mathrm{vec}\, Cz\mu^\top + \eta^{(2),\top}\,\mathrm{vec}\,\mu z^\top C^\top &= \eta^{(1),\top}Cz + 2\eta^{(2),\top}\,\mathrm{vec}\, Cz\mu^\top \\
&= \left(\eta^{(1),\top}C + 2\eta^{(2),\top}(\mu \otimes C)\right)z \tag{57} \\
&= \hat{\gamma}^{(1),\top}z,
\end{aligned}$$

i.e.,

$$\hat{\gamma}^{(1)} = C^\top \eta^{(1)} + 2(\mu \otimes C)^\top \eta^{(2)}. \tag{58}$$

Regrouping all the constants in (54), we obtain for $\hat{\gamma}^{(0)}$,

$$\hat{\gamma}^{(0)} = \eta^{(0)} + \eta^{(1),\top}\mu + \eta^{(2),\top}\,\mathrm{vec}\,\mu\mu^\top. \tag{59}$$

Now, we want to rewrite the regression on $s(Z)$ in terms of $t(Z)$ where

$$t(Z) = \left(1, Z^\top, \frac{Z_1^2 - 1}{\sqrt{2}}, Z_1 Z_2, \ldots, Z_1 Z_d, \frac{Z_2^2 - 1}{\sqrt{2}}, Z_2 Z_3, \ldots, \frac{Z_d^2 - 1}{\sqrt{2}}\right)^\top, \tag{60}$$

which satisfies $\mathbb{E}_Z\left[tt^\top\right] = I_{m'}$ with $m' = d + d(d+1)/2 + 1$. We do that in two steps, let

$$t_1(Z) = \left(1, Z^\top, \frac{Z_1^2 - 1}{\sqrt{2}}, Z_1 Z_2, \ldots, Z_1 Z_d, Z_1 Z_2, \frac{Z_2^2 - 1}{\sqrt{2}}, Z_2 Z_3, \ldots, \frac{Z_d^2 - 1}{\sqrt{2}}\right)^\top. \tag{61}$$

Let $\tilde{\gamma} = (\tilde{\gamma}^{(0)}, \hat{\gamma}^{(1),\top}, \tilde{\gamma}^{(2),\top})^\top \in \mathbb{R}^{1+d+d^2}$ be such that

$$\tilde{\gamma}^\top t_1(Z) = \hat{\gamma}^\top s(Z), \tag{62}$$

i.e., keeping only the constant terms and the terms quadratic in $Z$,

$$\hat{\gamma}^{(0)} + \sum_{j=1}^{d^2}\hat{\gamma}_j^{(2)}(\mathrm{vec}\, ZZ^\top)_j = \tilde{\gamma}^{(0)} + \sum_{k=0}^{d-1}\tilde{\gamma}_{1+(d+1)k}^{(2)}\left\{\frac{Z_k^2 - 1}{\sqrt{2}}\right\} + \sum_{j \neq 1+(d+1)k}\tilde{\gamma}_{2,j}(\mathrm{vec}\, ZZ^\top)_j. \tag{63}$$

We set, for any $k \geq 0$, $\tilde{\gamma}_{1+(d+1)k}^{(2)} = \hat{\gamma}_{1+(d+1)k}^{(2)}\sqrt{2}$, and $\tilde{\gamma}^{(0)} = \hat{\gamma}^{(0)} + \sum_{k=0}^{d-1}\hat{\gamma}_{(1+(d+1)k)}^{(2)}$. Then, (62) and (63) are satisfied. To go from $t_1$ to $t$, we need to get rid of the coordinates $t(Z)_k = Z_i Z_j$ for some $i > j$, i.e., $k \in [dp + 1, (d+1)p]$ for some integer $p$. Let $\Gamma = \mathrm{unvec}(\hat{\gamma}^{(2)})$, and let $\gamma^{(2)} \in \mathbb{R}^{d(d+1)/2}$ be defined by

$$\gamma^{(2)} = \left(\sqrt{2}\Gamma_{1,1}, 2\Gamma_{1,2}, \ldots, 2\Gamma_{1,d}, \sqrt{2}\Gamma_{2,2}, 2\Gamma_{2,3}, \ldots, 2\Gamma_{2,d}, \ldots, \sqrt{2}\Gamma_{d,d}\right)^\top. \tag{64}$$

Then $\gamma = [\tilde{\gamma}^{(0)}, \hat{\gamma}^{(1),\top}, \gamma^{(2),\top}]^\top \in \mathbb{R}^m$ satisfies

$$\gamma^\top t(Z) = \tilde{\gamma}^\top t_1(Z) = \hat{\gamma}^\top s(Z) = \eta^\top s(X). \tag{65}$$

All the previous computations give the expression of $\gamma$ as a function of $\eta$. $\qquad\square$

We now turn to prove Theorem 4.1 using the previous Lemma.

*Proof of Theorem 4.1.* As in the proof of Theorem D.2, the least squares regression on $s(X)$ can be rewritten in terms of $t(Z)$. Then, by applying Lemma D.3, we can map the regressor $\gamma$ with respect to $t$, to the regressor with respect to $s$, given $\beta = \phi(\eta)$. Since $\mathbb{E}_Z[tt^\top] = I$, the OLS simplifies to $\gamma = \mathbb{E}_Z[t(Z)f(\mu + CZ)]$. By Lemma D.3, the mapping from $\gamma$ to $\beta$ is given by

$$\beta = \begin{pmatrix} \gamma^{(0)} - \sum_{i=1}^d \Gamma_{i,i} - \beta^{(1),\top}\mu - \beta^{(2),\top}\,\mathrm{vec}\,\mu\mu^\top \\ C^{-\top}\gamma^{(1)} - 2\mu^\top \beta^{(2)} \\ \mathrm{vec}\left(C^{-1}\Gamma C^{-\top}\right) \end{pmatrix}, \tag{66}$$

where

$$\Gamma = \begin{pmatrix} \gamma_1^{(2)}/\sqrt{2} & \gamma_2^{(2)}/2 & \cdots & \cdots & \gamma_d^{(2)}/2 \\ \gamma_2^{(2)}/2 & \gamma_{d+1}^{(2)}/\sqrt{2} & \cdots & \cdots & \gamma_{2d-1}^{(2)}/2 \\ \vdots & & \ddots & & \vdots \\ \vdots & & & \ddots & \vdots \\ \gamma_d^{(2)}/2 & \cdots & \cdots & \cdots & \gamma_{d(d+1)/2}^{(2)}/\sqrt{2} \end{pmatrix}, \tag{67}$$

or componentwise $\Gamma_{i,i} = \gamma_{1+1/2(2d+2-i)(i-1)}^{(2)}/\sqrt{2}$, $\Gamma_{i,i+k} = \gamma_{1+1/2(2d+2-i)(i-1)+k}^{(2)}/2$ for $1 \leq i \leq d$ and $1 \leq k \leq d-i$, and $\Gamma_{i,j} = \Gamma_{j,i}$ for $j < i$. Regarding the complexity, the computation of the Cholesky matrix $C$ and its inverse requires $\mathcal{O}(d^3)$ operations; consequently, the computation of $\gamma$ can also be performed in $\mathcal{O}(d^3)$ operations. Using (66) to map $\gamma$ to $\eta$, involves computing $\mathrm{vec}\left(C^{-1}\Gamma C^{-\top}\right)$ and $C^{-\top}\gamma^{(1)}$, both of which can be performed in $\mathcal{O}(d^3)$. $\qquad \square$

### D.5 Concentration bounds for the Fisher matrix in the compact case

We now prove a Lemma to control the bias induced by inverting the estimated FIM, conditioned on the event that the estimated FIM is well-conditioned, which happens with high-probability given the number of samples $N$ is large enough. We first prove a version of the Lemma when $s$ is bounded (Lemma D.4), and then tackle the case where $s$ is unbounded but with bounded-moments (Lemma D.5).

**Lemma D.4** (Mean error bound for the inverse of $\hat{F}$ when $s$ is uniformly bounded). *Let $\delta \in (0,1)$, $N \geq B(4/3r + 2B)r^{-2}\log(2m\delta^{-1})$, $\omega \in \mathcal{W}$, and $\mathcal{A}(\omega) = [\|F_\omega - \hat{F}_\omega\| < \|F_\omega^{-1}\|^{-1}]$. Then, under Assumptions 2.1, 3.2, and $\|s\|_2^2 \leq B$, $\mathcal{A}(\omega)$ occurs with probability at least $1 - \delta$. Furthermore,*

$$\left\|\mathbb{E}\left[\hat{F}_\omega^{-1} - F_\omega^{-1}\Big|\mathcal{A}(\omega)\right]\right\| = \mathcal{O}(N^{-1}), \tag{68}$$

*where the constant in the big-$\mathcal{O}$ term can be chosen independently of $\omega$.*

*Proof of Lemma D.4 (exponential tail bound).* Fix $\omega \in \mathcal{W}$. For the sake of notation, we drop the subscript in $\omega$ but indicate the dependency in $N$. For any $N \geq 1$, let $\hat{F}_N = N^{-1}\sum_{i=1}^N ss^\top(X_i)$ with $X_1, \ldots, X_N \overset{\text{i.i.d}}{\sim} q$. Conditionally on $\mathcal{A}^{(N)} = [\|\hat{F}_N - F\| < \|F^{-1}\|^{-1}]$, $\hat{F}_N = F(I - (I - F^{-1}\hat{F}_N))$ is invertible because $F$ is invertible thanks to Assumption 2.1 and $0 < 1 - \|F^{-1}\|\|F - \hat{F}_N\| \leq 1 - \|I - F^{-1}\hat{F}_N\| = \|I - (I - F^{-1}\hat{F}_N)\|$. Using the Neumann series, we have

$$\hat{F}^{-1} - F^{-1} = \left(I - F^{-1}\hat{F} + \mathcal{O}(\|I - F^{-1}\hat{F}\|^2)\right)F^{-1}. \tag{69}$$

Since $I - F^{-1}\hat{F}_N$ is an average of $N$ i.i.d random matrices with uniformly bounded second moments thanks to Assumptions 3.1 and 3.2, $\{\sqrt{N}(I - F^{-1}\hat{F}_N)_{i,j}\}_{N\geq 1}$ is uniformly integrable (both in $N$ and $\omega$). By the strong law of large numbers, we have $\hat{F}_N \overset{a.s.}{\to} F$, thus $\mathbb{1}[\mathcal{A}^{(N)}] \overset{a.s.}{\to} 1$. Since $\{\sqrt{N}(I - F^{-1}\hat{F}_N)_{i,j}\}$ is uniformly integrable, the central limit also holds for the sequence of conditional random matrix components $\sqrt{N}(I - F^{-1}\hat{F}_N)_{i,j} \mid \mathcal{A}^{(N)}$. Then for any $1 \leq i, j \leq m$, $\sqrt{N}(I - F^{-1}\hat{F})_{i,j} \mid \mathcal{A}^{(N)}$ converges in law with uniformly bounded variance. Thus, conditioned on $\mathcal{A}^{(N)}$, $\sqrt{N}(I - F^{-1}\hat{F})_{i,j} = \mathcal{O}_P(1)$, which implies that $\sqrt{N}\|I - F^{-1}\hat{F}\|_F = \mathcal{O}_P(1)$. Thus $N\|I - F^{-1}\hat{F}_N\|^2 \mid \mathcal{A}^{(N)} = \mathcal{O}_P(1)$, and taking the expectation in (69) yields $\|\mathbb{E}[\hat{F}^{-1} - F^{-1} \mid \mathcal{A}^{(N)}]\| = \mathcal{O}(N^{-1})$, again using Assumption 3.2.

By [57, Th. 1.62] with uniform bound $\|(ss^\top(X_i) - F)/N\| \leq 2B/N$ and variance $\|\sum_{i=1}^N \mathbb{E}[((ss^\top(X_i) - F)/n)^2]\| \leq B\|F\|/N$, and the definition of $r$ (Assumption 3.2), we have

$$P(\|\hat{F}_N - F\| \geq \|F^{-1}\|^{-1}) \leq P(\|\hat{F}_N - F\| \geq r)$$
$$\leq 2m\exp\left(-\frac{Nr^2}{B(4/3r + 2\|F\|)}\right) \tag{70}$$
$$\leq 2m\exp\left(-\frac{Nr^2}{B(4/3r + 2B)}\right),$$

where to go from the second to the third line, we use $\|F\| \leq B$. Setting $N \geq B(4/3r + 2B)r^{-2} \log(2m\delta^{-1})$ yields $P(\|\hat{F}_N - F\| \geq \|F^{-1}\|^{-1}) \leq \delta$, i.e., $P(\mathcal{A}) \geq 1 - \delta$. The bound is independent of $\omega$, and true for any $\omega \in \mathcal{W}$, finally yielding the result. $\qquad\square$

**Lemma D.5.** *Under Assumptions 2.1, 3.1 and 3.2, for* $\sqrt{N} \geq r^{-2}\delta^{-1}(\sqrt{8e \log(m)}\mu_4\nu + 8e\mu_4^2 \log(m))$, $P(\mathcal{A}(\omega)) \geq 1 - \delta$ *and* (68) *holds.*

*Proof of Lemma D.5 (polynomial tail bound).* We follow the proof of D.4. The CLT for $\hat{F}$ still holds thanks to Assumption 3.1, and by the same argument as in D.4, the conditional CLT is still valid. Thus, $\|\mathbb{E}[I - F^{-1}F \mid \mathcal{A}]\| = \mathcal{O}(N^{-1})$, and the constant inside the big-$\mathcal{O}$ notation is also independent on $\omega$ using the uniform bounds on the fourth-moment of $s$. By the definition of $r$, the Bienaymé-Tchebychev's inequality, and [38, Theorem 3.1], we have

$$
\begin{aligned}
P(\|\hat{F}_N - F\| \geq \|F^{-1}\|^{-1}) &\leq P(\|\hat{F}_N - F\| \geq r) \\
&\leq \mathbb{E}\|\hat{F}_N - F\|^2 / r^2 \\
&\leq r^{-2} \left\{ \sqrt{8e \frac{\log(m)}{N}} \mu_4 \nu + 8e\mu_4^2 \frac{\log(m)}{N} \right\} \\
&\leq \frac{r^{-2}}{\sqrt{N}} \left\{ \sqrt{8e \log(m)} \mu_4 \nu + 8e\mu_4^2 \log(m) \right\},
\end{aligned}
\tag{71}
$$

where last line is solely here to simplify the requirement on $N$. Setting $\sqrt{N} \geq r^{-2}\delta^{-1}(\sqrt{8e \log(m)}\mu_4\nu + 8e\mu_4^2 \log(m))$ yields $P(\|\hat{F}_N - F\| \geq \|F^{-1}\|^{-1}) \leq \delta$, i.e., $P(\mathcal{A}) \geq 1 - \delta$. The bound is independent of $\omega$, and true for any $\omega \in \mathcal{W}$. $\qquad\square$

### D.6 Convergence analysis of the stochastic LSVI algorithm

The proof of Theorem 3.4 relies on Lemmas D.6, D.7, D.9, and D.10. Lemma D.6 states the equivalence between stochastic mirror descent and stochastic natural gradient descent, the proof is very similar to the non-stochastic case (Proposition 2.4). Lemma D.7 gives the general convergence rate for stochastic mirror descent with the presence of an additional bias under the assumption the bias has bounded variance. This is a generalisation of Hanzely and Richtárik [30, Th. 4.5]. Both Lemmas D.9, D.10 are required to handle the two first moments of the bias induced by inverting the FIM estimate. This analysis requires conditioning on the event that the estimated FIMs are well-conditioned, which happens with high-probability (Lemma D.5). Theorem 3.4 follows by successively applying Lemma D.6 and Lemma D.7, the latter requires Lemmas D.9 and D.10.

**Lemma D.6** (Equivalence between stochastic mirror descent and stochastic natural gradient descent). *Define the stochastic gradient $\hat{\nabla}_\omega l$ by*

$$
\hat{\nabla}_\omega l : \omega \mapsto \eta(\omega) - \hat{F}_\omega^{-1} \hat{z}_\omega,
\tag{72}
$$

*given that $\hat{F}_\omega$ is invertible. Then* (10) *is equivalent to*

$$
\hat{\eta}_{t+1} = \hat{\eta}_t - \varepsilon_t \hat{\nabla}_\omega l(\hat{\omega}_t),
\tag{73}
$$

*where $\hat{\omega}_t = \omega(\hat{\eta}_t)$. Furthermore, the previous dynamic is equivalent to*

$$
\hat{\omega}_{t+1} = \operatorname{argmin}_{\omega \in \mathcal{W}} \left\{ \hat{\nabla}_\omega^\top l(\hat{\omega}_t)\omega + \frac{1}{\varepsilon_t} D_{Z^*}(\omega, \hat{\omega}_t) \right\},
\tag{74}
$$

*with $\hat{\eta}_{t+1} = \eta(\hat{\omega}_{t+1})$.*

*Proof.* The first equivalence follows from the same computations as in Proposition 2.4. Let us show that iteration (10) can be recovered as the dual in the natural parameter space of a stochastic mirror descent, i.e., that $\hat{\eta}_{t+1} = \eta(\hat{\omega}_{t+1})$ with $(\hat{\omega}_t)$ given by (74) recovers (10). The first order condition on (74) gives

$$
\nabla Z^*(\hat{\omega}_{t+1}) = \nabla Z^*(\hat{\omega}_t) - \varepsilon_t \hat{\nabla}_\omega l(\hat{\omega}_t).
\tag{75}
$$

However, since $\nabla Z^*(\hat{\omega}_{t+1}) = \eta(\hat{\omega}_{t+1}) = \hat{\eta}_{t+1}$, the desired equivalence between the two dynamics follows. $\qquad\square$

**Lemma D.7** (General convergence for biased stochastic mirror descent). *Let us define the bias $B_t$ of the stochastic gradient at iteration $t$ by*

$$B_t = \mathbb{E}[\hat{\nabla}_\omega l(\hat{\omega}_t) - \nabla_\omega l(\hat{\omega}_t) \mid \hat{\omega}_t], \tag{76}$$

*given that $\hat{F}_{\hat{\omega}_t}$ is invertible, and let us denote by $m(\hat{\omega}_t) := \omega_{t+1,*}$ the exact mirror-descent iterate starting from $\hat{\omega}_t$, i.e.,*

$$\omega_{t+1,*} = \operatorname{argmin}_{\omega \in \mathcal{W}} \left\{ \nabla_\omega^\top l(\hat{\omega}_t)\omega + \varepsilon_t^{-1} D_{Z^*}(\omega, \hat{\omega}_t) \right\}. \tag{77}$$

*Assume there exists $\sigma^2 > 0$ (to be specified later) such that for any $t \geq 0$,*

$$\mathbb{E}\left[B_t^\top (\omega_{t+1,*} - \hat{\omega}_{t+1}) \big| \hat{\omega}_t\right] \leq \sigma^2 \varepsilon_t. \tag{78}$$

*Let $\varepsilon_t \leq \frac{1}{L} \wedge \frac{1}{\mu}$ for all $t \geq 0$, let $c_t = c_{t-1}\varepsilon_{t-1}^{-1}(\varepsilon_t^{-1} - \mu)^{-1}$ for $t \geq 1$, and let $c_0 = 1$. Let $C_k = \sum_{t=1}^k c_{t-1}$ for $k \geq 1$. Then, under Assumptions 2.1, 2.5, and the additional bounded-noise assumption (78),*

$$\frac{1}{C_k} \sum_{t=1}^k c_{t-1}\mathbb{E}[l(\hat{\omega}_t) - l(\omega^*)] \leq \frac{(\varepsilon_0^{-1} - \mu)\, \mathrm{uKL}(q_{\omega^*} \mid q_{\omega_0})}{C_k} + \sigma^2 \sum_{t=0}^{k-1} \frac{c_t \varepsilon_t}{C_k}$$
$$+ \sum_{t=0}^{k-1} \frac{c_t}{C_k} \mathbb{E}[B_t^\top(\omega^* - \hat{\omega}_{t+1})]. \tag{79}$$

*Proof.* Assumption 2.1 allows us to define (77). Under Assumption 2.5 and the boundedness of the gradient estimate (78), we can derive a slightly modified version of the descent lemma [30, Lemma 4.3] which accounts for the presence of the bias. Next line follows from the calculations done in the proof of Hanzely and Richtárik [30, Lemma 4.3]:

$$\mathbb{E}[l(\hat{\omega}_{t+1}) - l(\omega^*) \mid \hat{\omega}_t] \leq \left(\frac{1}{\varepsilon_t} - \mu\right) D_{Z^*}(\omega^*, \hat{\omega}_t) - \frac{1}{\varepsilon_t}\mathbb{E}[D_{Z^*}(\omega^*, \hat{\omega}_{t+1}) \mid \hat{\omega}_t]$$
$$+ \varepsilon_t \sigma^2 - \left(\frac{1}{\varepsilon_t} - L\right) \mathbb{E}[D_{Z^*}(\hat{\omega}_{t+1}, \hat{\omega}_t) \mid \hat{\omega}_t] + B_t^\top(\omega^* - \omega_{t+1,*}) \tag{80}$$
$$- \mathbb{E}[B_t^\top(\hat{\omega}_{t+1} - \omega_{t+1,*}) \mid \hat{\omega}_t],$$

where $\omega^* = \operatorname{argmin}_{\omega \in \mathcal{W}} l(\omega)$. Since $\varepsilon_t^{-1} \geq L$, the fourth term is negative. Therefore, (80) becomes

$$\mathbb{E}[l(\hat{\omega}_{t+1}) - l(\omega^*) \mid \hat{\omega}_t] \leq \left(\frac{1}{\varepsilon_t} - \mu\right) D_{Z^*}(\omega^*, \hat{\omega}_t) - \frac{1}{\varepsilon_t}\mathbb{E}[D_{Z^*}(\omega^*, \hat{\omega}_{t+1}) \mid \hat{\omega}_t]$$
$$+ \varepsilon_t \sigma^2 + B_t^\top(\omega^* - \omega_{t+1,*}) - \mathbb{E}[B_t^\top(\hat{\omega}_{t+1} - \omega_{t+1,*}) \mid \hat{\omega}_t]. \tag{81}$$

Taking the expectation of (81) gives

$$\mathbb{E}[l(\hat{\omega}_{t+1}) - l(\omega^*)] \leq \left(\frac{1}{\varepsilon_t} - \mu\right) \mathbb{E}[D_{Z^*}(\omega^*, \hat{\omega}_t)] - \frac{1}{\varepsilon_t}\mathbb{E}[D_{Z^*}(\omega^*, \hat{\omega}_{t+1})]$$
$$+ \varepsilon_t \sigma^2 + \mathbb{E}[B_t^\top(\omega^* - \hat{\omega}_{t+1})]. \tag{82}$$

Let $c_t = c_{t-1}\varepsilon_{t-1}^{-1}(\varepsilon_t^{-1} - \mu)^{-1}$ for $t \geq 1$, and let $c_0 = 1$. Let $k \geq 1$ and define $C_k = \sum_{t=1}^k c_{t-1}$. Since $\varepsilon_t \leq \frac{1}{\mu}$, we have $c_t \geq 0$. Multiply by $c_t \geq 0$ (82) and sum for $t \in [1, k]$, then divide by $C_k$,

$$\sum_{t=1}^k \frac{c_{t-1}}{C_k}\mathbb{E}[l(\hat{\omega}_t) - l(\omega^*)] \leq \frac{(\varepsilon_0^{-1} - \mu)D_{Z^*}(\omega^*, \omega_0)}{C_k} + \sigma^2 \sum_{t=0}^{k-1} \frac{c_t \varepsilon_t}{C_k} + \sum_{t=0}^{k-1} \frac{c_t}{C_k}\mathbb{E}[B_t^\top, (\omega^* - \hat{\omega}_{t+1})]. \tag{83}$$

We essentially recover Hanzely and Richtárik [30, Th. 4.5], but with the additional bias terms. Finally, (79) follows from (83) and $D_{Z^*}(\omega^*, \omega_0) = \mathrm{uKL}(q_{\omega^*} \mid q_{\omega_0})$. □

*Remark D.8.* The previous lemma requires a boundedness assumption on the gradient estimate given by (78). This assumption is typically required for proving such descent lemmas, see [20, 30, 36, 37, 58]. In particular, this assumption is implied, by Cauchy-Schwarz inequality, if the gradient estimate $\hat{\nabla}_\omega l$ mapping given by (72) has bounded bias and variance, or directly if the gradient estimate is unbiased. We will prove it is satisfied under Assumptions 3.1 and 3.3 in Lemma D.10.

**Lemma D.9** (Controlling the bias terms in (79)). *Let $k \geq 0$, and let $\mathcal{A}_k = \cap_{t=0}^k \mathcal{A}(\hat{\omega}_t)$ with $\mathcal{A}(\omega) = [\|F_\omega - \hat{F}_\omega\| < \|F_\omega^{-1}\|^{-1}]$, then under Assumptions 2.1, 3.1, 3.2 and 3.3, for any $t \in [0, k]$*

$$\mathbb{E}[B_t^\top (\omega^* - \hat{\omega}_{t+1}) \mid \mathcal{A}_k] \leq \mathcal{O}(N^{-1}) \times m_2 m^{1/2} \mu_4 \times \left(\mathbb{E}[\|\omega^* - \hat{\omega}_{t+1}\|^2 \mid \mathcal{A}_k]\right)^{1/2}. \quad (84)$$

*Proof.* By Cauchy Schwarz inequality, for any $t \in [0, k]$,

$$\mathbb{E}[B_t^\top (\omega^* - \hat{\omega}_{t+1}) \mid \mathcal{A}_k] \leq \left(\mathbb{E}[\|B_t\|^2 \mid \mathcal{A}_k]\mathbb{E}[\|\omega^* - \hat{\omega}_{t+1}\|^2 \mid \mathcal{A}_k]\right)^{1/2}. \quad (85)$$

Conditionally on $\mathcal{A}_k$, $\hat{F}_{\hat{\omega}_t}$ is invertible, and by Lemma D.5, there exists $C > 0$ such that for $N$ large enough, $N\|\mathbb{E}[F_{\hat{\omega}_t}^{-1} - \hat{F}_{\hat{\omega}_t}^{-1} \mid \mathcal{A}_k, \hat{\omega}_t]\| \leq C$. Consequently,

$$
\begin{aligned}
N^2 \|B_t\|^2 &= N^2 \|\mathbb{E}[F_{\hat{\omega}_t}^{-1} z_{\hat{\omega}_t} - \hat{F}_{\hat{\omega}_t}^{-1} \hat{z}_{\hat{\omega}_t} \mid \mathcal{A}_k, \hat{\omega}_t]\|^2 \\
&= N^2 \|\mathbb{E}[F_{\hat{\omega}_t}^{-1} - \hat{F}_{\hat{\omega}_t}^{-1} \mid \mathcal{A}_k, \hat{\omega}_t] z_{\hat{\omega}_t}\|^2 \\
&\leq N^2 \|\mathbb{E}[F_{\hat{\omega}_t}^{-1} - \hat{F}_{\hat{\omega}_t}^{-1} \mid \mathcal{A}_k, \hat{\omega}_t]\|^2 \|z_{\hat{\omega}_t}\|^2 \\
&\leq C^2 \|z_{\hat{\omega}_t}\|^2 \\
&\leq C^2 m_2^2 m \mu_4^2,
\end{aligned}
\quad (86)
$$

where we used to go from the first to the second line, the independency of $z_{\hat{\omega}_t}$ with $\hat{F}_{\hat{\omega}_t}$ conditioned on $\hat{\omega}_t$ and $\mathbb{E}[\hat{z}_{\hat{\omega}_t} \mid \hat{\omega}_t] = z_{\hat{\omega}_t}$, and to go from the third to the fourth line, we use Lemma D.5, requiring Assumptions 2.1, 3.1 and 3.2, and to go from the fourth to the fifth line, we use bounds on the moment of $s^2$ and $f^2$ given by Assumptions 3.1, 3.3:

$$
\begin{aligned}
\|z_\omega\| &\leq (\mathbb{E}\|s\|^2)^{1/2}(\mathbb{E}f^2)^{1/2} \\
&\leq \left(m \sup_{\omega \in \mathcal{W}} \max_{1 \leq i \leq m} \mathbb{E}|s|_i^2\right)^{1/2} \times m_2 \\
&\leq m^{1/2} \mu_4 m_2.
\end{aligned}
\quad (87)
$$

Taking the expectation of (86) conditioned on $\mathcal{A}_k$ and plugging it into (85) yields (84). $\quad\square$

**Lemma D.10** (High-probability uniform bound for the variance of the gradient estimate). *Let $\varepsilon > 0$. For any $\omega \in \mathcal{W}$, let $M(\omega) = \operatorname{argmin}_{\omega' \in \mathcal{W}}\{\nabla_\omega^\top l(\omega)\omega' + \varepsilon D_{Z^*}(\omega', \omega)\}$ be the exact mirror-descent iterate starting from $\omega$ with step size $\varepsilon$. Similarly, let $\hat{M}(\omega)$ be the mirror-descent using gradient estimate $\hat{\nabla}_\omega l$ given by (72). Let $\sigma^2(\omega)$ be defined by*

$$\sigma^2(\omega) = \varepsilon^{-1}\mathbb{E}[(\mathbb{E}[\hat{\nabla}_\omega l(\omega)] - \hat{\nabla}_\omega l(\omega))^\top (M(\omega) - \hat{M}(\omega))], \quad (88)$$

*where all the expectations are taken conditioned on $\mathcal{A}(\omega)$. Under Assumptions 2.1, 3.1, and 3.3, there exists some constant $C > 0$, such that for $N$ large enough (see Lemma D.5), and any $\omega \in \mathcal{W}$,*

$$\sigma^2(\omega) \leq N^{-1} Z_\omega (C + o_\varepsilon(1)), \quad (89)$$

*for some constant $C > 0$, and the little-o term is independent of $\omega, N$.*

*Proof.* To obtain the uniform bound on (88), we independently bound both terms in the scalar product. Let $\omega \in \mathcal{W}$, and let $\eta$ be the corresponding natural parameter, $\eta = \eta(\omega)$. Conditionally on $\mathcal{A}(\omega)$, by the computations done in the proof of Lemmas D.4 and D.5 there exists a constant $C > 0$ such that for $N$ large enough, $N\mathbb{E}[\|\hat{F}_\omega^{-1} - F_\omega^{-1}\|^2] \leq C$, i.e., the mean-square error is $\mathcal{O}(1/N)$, and thus the variance $\mathbb{E}[\|\hat{F}_\omega^{-1} - \mathbb{E}[\hat{F}_\omega^{-1}]\|^2]$ is $\mathcal{O}(1/N)$.

Using the definition of the stochastic gradient $\hat{\nabla} l$ given by (72), and the previous bound, the first term is bounded by:

$$
\begin{aligned}
N\mathbb{E}[\|\mathbb{E}[\hat{\nabla}_\omega l(\omega)] - \hat{\nabla}_\omega l\|^2] &= N\mathbb{E}\left[\|(\mathbb{E}[\hat{F}_\omega^{-1}] - \hat{F}_\omega^{-1})z_\omega + \hat{F}_\omega^{-1}(z_\omega - \hat{z}_\omega)\|^2\right] \\
&\leq 2N\left(\mathbb{E}[\|(\mathbb{E}[\hat{F}_\omega^{-1}] - \hat{F}_\omega^{-1})z_\omega\|^2] + \mathbb{E}[\|\hat{F}_\omega^{-1}(z_\omega - \hat{z}_\omega)\|^2]\right) \\
&\leq 2N\left(\|z_\omega\|^2 \times \mathbb{E}[\|(\mathbb{E}[\hat{F}_\omega^{-1}] - \hat{F}_\omega^{-1})\|^2]\right. \\
&\quad \left. + \mathbb{E}[\|\hat{F}_\omega^{-1}(z_\omega - \hat{z}_\omega)\|^2]\right) \\
&\leq 2N\left(m\mu_4^2 m_2^2 \times C/N + \mathbb{E}\|\hat{F}_\omega^{-1}(z_\omega - \hat{z}_\omega)\|^2\right),
\end{aligned}
\quad (90)
$$

where we used $(a+b)^2 \leq 2a^2 + 2b^2$ from the first to the second line, from the second to the third line the sub-multiplicativity of $\|\cdot\|$, from the third to the fourth $\|z_\omega\|^2 \leq m\mu_4^2 m_2^2$ and the $\mathcal{O}(1/N)$ bound on the variance of $\hat{F}_\omega^{-1}$. Furthermore, we can bound the last term in (90) by

$$
\begin{aligned}
N\mathbb{E}[\|\hat{F}_\omega^{-1}(z_\omega - \hat{z}_\omega)\|^2] &\leq N\mathbb{E}[\|\hat{F}_\omega^{-1}\|^2]\mathbb{E}[\|z_\omega - \hat{z}_\omega\|^2] \\
&\leq 2N\{\mathbb{E}[\|\hat{F}_\omega^{-1} - F_\omega^{-1}\|^2 + \mathbb{E}\|F_\omega^{-1}\|^2]\}\mathbb{E}[\|z_\omega - \hat{z}_\omega\|^2] \\
&\leq 2N(r^{-2} + C/N) \times \mathbb{E}[\|z_\omega - \hat{z}_\omega\|^2] \\
&\leq 2N(r^{-2} + C/N) \times C'/N
\end{aligned}
\tag{91}
$$

where we used that $r^{-2} = \sup_\omega \|F_\omega^{-1}\|^2$, and $\mathbb{E}[\|z_\omega - \hat{z}_\omega\|^2] = \mathrm{Tr}(\mathrm{Cov}(sf(X)))/N$ with Assumptions 3.1 and 3.3 to state that there exists $C' > 0$ independent on $\omega$ such that $\mathbb{E}[\|z_\omega - \hat{z}_\omega\|^2] \leq C'/N$. Gathering (90) and (91) yields for the first term of the scalar product:

$$
N\mathbb{E}[\|\mathbb{E}[\hat{\nabla}_\omega l(\omega)] - \hat{\nabla}_\omega l\|^2] \leq 2Cm\mu_4^2 m_2^2 + 4C'(r^{-2} + C/N), \tag{92}
$$

which in turn can be bounded by some constant $C'' > 0$ independent of $\omega$. Let us tackle the second term inside the scalar product. By Proposition 2.4,

$$
M(\omega) = \omega(\eta - \varepsilon\nabla_\omega l(\omega)), \tag{93}
$$

and similarly for $\hat{M}$,

$$
\hat{M}(\omega) = \omega(\eta - \varepsilon\hat{\nabla}_\omega l(\omega)). \tag{94}
$$

Under Assumption 2.1, the mapping $\omega : \eta \in \mathcal{V} \mapsto \omega(\eta)$ is differentiable with $\nabla_\eta \omega = Z_\eta F_\eta = \int ss^\top q_\eta$, see (35). Let us denote by $H_i = D_\eta^2 \omega^{(i)}$ the Hessian of the $i$-th component application of $\omega$ for any $1 \leq i \leq m$, which is a $\mathbb{R}^{m \times m}$ matrix given by $D_\eta^2 \omega^{(i)} = \int s_i ss^\top q_\eta$, and let $D_\eta^2 \omega = (H_1, H_2, \ldots, H_m)^\top$ be the collection of the Hessian matrices. For any $h \in \mathbb{R}^m$, let us denote by $D_\eta^2 \omega[h,h] = D_\eta^2 \omega[h]^2 = (h^\top H_1 h, \ldots, h^\top H_m h)^\top \in \mathbb{R}^m$. A Taylor expansion with Lagrange remainder yields,

$$
\begin{aligned}
M(\omega) - \hat{M}(\omega) &= \omega(\eta - \varepsilon\nabla_\omega l(\omega)) - \omega(\eta - \varepsilon\hat{\nabla}_\omega l(\omega)) \\
&= \varepsilon(\nabla_\eta \omega)^\top(\hat{\nabla}_\omega l(\omega) - \nabla_\omega l(\omega)) + \varepsilon^2 \int_0^1 (1-t)\left\{D_\eta^2 \omega(\eta - t\varepsilon\nabla_\omega l(\omega))[\nabla_\omega l(\omega)]^2\right. \\
&\quad \left. - D_\eta^2 \omega(\eta - t\varepsilon\hat{\nabla}_\omega l(\omega))[\hat{\nabla}_\omega l(\omega)]^2\right\} \mathrm{d}t
\end{aligned}
\tag{95}
$$

Let $\hat{R}$ be the $\varepsilon^2$ remainder term in (95). Then, $\hat{R}/Z_\omega$ is a $\mathbb{R}^m$ vector whose norm can be uniformly bounded using the uniform bounds on the fourth-moment of $s$ using similar techniques as for the bound on $\|F_\omega\|$ (see below), we omit the details. This implies that $\varepsilon^2 \hat{R} = Z_\omega o(\varepsilon)$ with constant in the little-$o$ terms independent on $\omega$. Consequently,

$$
\begin{aligned}
\sqrt{N}(\mathbb{E}[\|M(\omega) - \hat{M}(\omega)\|^2])^{1/2} &= \sqrt{N}(\mathbb{E}[\|\varepsilon Z_\omega F_\omega(\hat{\nabla}_\omega l(\omega) - \nabla_\omega l(\omega)) + Z_\omega o(\varepsilon)\|^2])^{1/2} \\
&\leq 2\sqrt{N}\varepsilon Z_\omega \|F_\omega\|(\mathbb{E}[\|\hat{\nabla}_\omega l(\omega) - \nabla_\omega l(\omega)\|^2])^{1/2} + Z_\omega o(\varepsilon) \\
&\leq 2\varepsilon Z_\omega(\|F_\omega\|C^{(3)} + o(1)) \\
&\leq 2\varepsilon Z_\omega(m\mu_4^2 C^{(3)} + o(1)) \\
&\leq \varepsilon Z_\omega(C^{(4)} + o(1)),
\end{aligned}
\tag{96}
$$

for some constant $C^{(4)} > 0$, and where we used $\|F_\omega\|^2 \leq m^2\mu_4^4$:

$$
\begin{aligned}
\|F_\omega\|^2 &\leq \|F_\omega\|_F^2 \\
&\leq \sum_{1 \leq i,j \leq m} \sqrt{\mathbb{E}s_i^4 \mathbb{E}s_j^4} \\
&\leq m^2\mu_4^4,
\end{aligned}
\tag{97}
$$

using the definition of the Frobesnius norm and then Cauchy Schwarz inequality to bound componentwise $F_\omega$. By Cauchy Schwarz inequality, (92) and (96), for $N$ large enough,

$$
\begin{aligned}
N\sigma^2(\omega) &\leq \varepsilon^{-1}N(\mathbb{E}[\|\mathbb{E}[\hat{\nabla}_\omega l(\omega)] - \hat{\nabla}_\omega l(\omega)\|^2]\mathbb{E}[\|M(\omega) - \hat{M}(\omega)\|^2])^{1/2} \\
&\leq (C^{(4)}\sqrt{C''} + o(1))Z_\omega.
\end{aligned}
\tag{98}
$$

This concludes the proof. $\qquad\square$

With previous Lemmas D.6, D.7, D.9, and D.10 in hand, we can prove the main result.

*Proof of Theorem 3.4.* Define $\bar{\omega}_k$ as given in the theorem. By convexity of $l$,

$$l(\bar{\omega}_k) - l(\omega^*) \leq \frac{1}{C_k} \sum_{t=1}^{k} c_{t-1}(l(\hat{\omega}_t) - l(\omega^*)). \tag{99}$$

Combining Lemma D.7 with Lemma D.9 to control the bias terms, we find that the expectation of the RHS in (99) is upper bounded by

$$\mathbb{E}[l(\bar{\omega}_k) - l(\omega^*) \mid \mathcal{A}_k] \leq \frac{(\varepsilon_0^{-1} - \mu)\,\mathrm{uKL}(q_{\omega^*} \mid q_{\omega_0})}{C_k}$$
$$+ \sigma^2 \sum_{t=0}^{k-1} \frac{c_t \varepsilon_t}{C_k} + \mathcal{O}(N^{-1}) \times S_{k,N}, \tag{100}$$

where $S_{k,N} := \frac{m_2 m^{1/2} \mu_4}{C_k} \sum_{t=0}^{k-1} c_t (\mathbb{E}[\|\omega^* - \hat{\omega}_{t+1}\|^2 \mid \mathcal{A}_k])^{1/2}$, where the big-$\mathcal{O}$ term is independent of $k$ since it is independent of $\omega_0 = \hat{\omega}_0, \hat{\omega}_1, \ldots \hat{\omega}_k$, and $\sigma^2$ some upper bound of $\sup_{k \geq 1} \sigma^2(k)$ with $\sigma^2(k)$ satisfying the assumption of Lemma D.7, for all $t \leq k$:

$$\mathbb{E}\left[(\mathbb{E}[\hat{\nabla}_\omega l(\hat{\omega}_t) \mid \hat{\omega}_t, \mathcal{A}_k])^\top (\omega_{t+1,*} - \hat{\omega}_{t+1}) \Big| \hat{\omega}_t\right] \leq \sigma^2(k)\varepsilon_t. \tag{101}$$

By Lemma D.10, we can set

$$\sigma^2(k) = N^{-1} C \max_{0 \leq t \leq k-1} Z_{\hat{\omega}_t}, \tag{102}$$

for some constant $C$ independent on $N$ and the sequence $\hat{\omega}_0, \ldots, \hat{\omega}_{k-1}$.

Let us tackle the terms which depend both upon $N$ and $k$ via the sequence $\hat{\omega}_0, \hat{\omega}_1, \ldots, \hat{\omega}_k$. By the law of large numbers, as $N \to \infty$, $\hat{F}_{\omega_0} \to F_{\omega_0}$ and $\hat{z}_{\omega_0} \to z_{\omega_0}$ almost surely. Then, by the continuous mapping theorem, $\hat{\nabla}_\omega l(\omega_0) \to \nabla_\omega l(\omega_0)$ almost surely, and thus $\omega_1 \to \omega_{1,*} = \omega_1^*$ almost surely, where $\omega_1^*$ is the first mirror-descent iterate. By induction, we obtain that for any $k \geq 1$, $\hat{\omega}_t \to \omega_t^*$ a-s for all $t \in [1, k]$, i.e., the finite sequence $\{\omega_0, \ldots, \hat{\omega}_k\}$ converges to the exact mirror-descent sequence $\{\omega_0, \omega_1^*, \ldots, \omega_k^*\}$. We deduce that, almost surely, for all $t \geq 1$, $\|\omega^* - \hat{\omega}_t\|^2 \to \|\omega^* - \omega_t^*\|^2$ since the countable intersection of almost sure events is an almost sure event. By Aubin-Frankowski et al. [Th. 4 31], we know that the Mirror-Descent sequence $l(\omega_t^*)$ converges to $l(\omega^*)$. Since $l$ is strongly-convex, $l(\omega_t^*) \to l(\omega^*)$ implies that $\|\omega^* - \omega_t^*\| \to 0$ as $t$ goes to $\infty$. Combining with the previous almost-sure convergence, we obtain that for any $k \geq 1$, the following equality holds almost surely,

$$\lim_{N \to \infty} \max_{1 \leq t \leq k} \|\omega^* - \hat{\omega}_t\|^2 = \max_{1 \leq t \leq k} \|\omega^* - \omega_t^*\|^2 := D_k, \tag{103}$$

with $\sup_{k \geq 1} D_k < \infty$. For $k \geq 1$, let $U_k \subset \mathcal{W}$ be the closed-ball of center $\omega^*$ and of radius $2 \times D_k$, let $U_0 = \{\omega_0\}$, and let $U$ be the reunion of $U_0$ and the ball centered at $\omega^*$ with radius $\sup_{k \geq 1} D_k < \infty$. For each fixed $k \geq 1$, almost surely, when $N \to \infty$, for any $0 \leq t \leq k$, $\hat{\omega}_t \in U_k$, and therefore

$$\omega_0, \hat{\omega}_1, \hat{\omega}_2, \ldots, \hat{\omega}_k \in \cup_{k \geq 0} U_k \subset U. \tag{104}$$

Since $\omega \mapsto Z_\omega$ is continuous and $U$ is compact, we have $\sup_{\omega \in U} Z_\omega < \infty$. Then almost surely, as $N \to \infty$, $\sup_{k \geq 1} \sigma^2(k) \to N^{-1} C \sup_{\omega \in U} Z_\omega := \sigma^2$, and therefore almost surely, as $N \to \infty$,

$$\sup_{k \geq 1} \sigma^2(k) \leq 2N^{-1} C \sup_{\omega \in U} Z_\omega = 2\sigma^2 < \infty. \tag{105}$$

Almost surely, when $N \to \infty$, for any $0 \leq t \leq k$, $\|\omega^* - \hat{\omega}_t\|^2 \leq 2D_k$, which implies that $\sup_{0 \leq t \leq k-1} \mathbb{E}[\|\omega^* - \hat{\omega}_{t+1}\|^2 \mid \mathcal{A}_k]^{1/2} < (2 \sup_{k \geq 1} D_k)^{1/2} < \infty$. Using $\sum_{t=0}^{k-1} c_t = C_k$, and bounding uniformly the summands of $S_{k,N}$ yields

$$S_{k,N} \leq m_2 m^{1/2} \mu_4 (2 \sup_{k \geq 1} D_k)^{1/2}. \tag{106}$$

Finally, plugging (105) and (106) into (100) yields the uniform bound over $k$:

$$\mathbb{E}[l(\bar{\omega}_k) - l(\omega^*) \mid \mathcal{A}_k] \leq \frac{(\varepsilon_0^{-1} - \mu)\,\mathrm{uKL}(q_{\omega^*} \mid q_{\omega_0})}{C_k} + \mathcal{O}(N^{-1}) \sum_{t=0}^{k-1} \frac{c_t \varepsilon_t}{C_k} + \mathcal{O}(N^{-1}). \quad (107)$$

All the constants in the big-$\mathcal{O}$ terms can be chosen independently on the sequence of $\hat{\omega}$.

Using Proposition D.5 with $\delta/(k+1)$ and a union bound, we have $P(\cap_{t=0}^k \mathcal{A}(\hat{\omega}_t)) \geq 1 - \delta$ for the chosen $N$.

Finally, let us prove the explicit convergence rates for linearly increasing stepsizes $\varepsilon_t = (L + \alpha t)^{-1}$, $t \geq 0$. Similarly to Hanzely and Richtárik [Lemma 4.8 30], we distinguish three cases depending on $\alpha$ compared to $\mu$. If $\alpha > \mu$, then $C_k = \Omega(k^{\mu/\alpha})$ and $\sum_{t=0}^{k-1} c_t \varepsilon_t = \mathcal{O}(1)$ which yields $\mathcal{O}(k^{-\mu/\alpha}) + \mathcal{O}(N^{-1})$ for the RHS of (12). If $\alpha = \mu$, then $C_k = \Omega(k)$ and $\sum_{t=0}^{k-1} c_t \varepsilon_t = O(\log(k))$. If $\alpha < \mu$, then $C_k = \Omega(k^{\mu/\alpha})$ and $\sum_{t=0}^{k-1} c_t \varepsilon_t = \mathcal{O}(k^{\mu/\alpha-1})$. $\qquad \square$

# NeurIPS Paper Checklist

1. **Claims**

   Question: Do the main claims made in the abstract and introduction accurately reflect the paper's contributions and scope?

   Answer: [Yes]

   Justification: The main contributions of the paper are as follows: (i) KL minimisation within exponential families can be performed via successive linear regressions under tempered variational approximations. This approach is equivalent to natural gradient descent (NGD) and mirror descent (MD) but avoids the need for explicit gradient-based procedures. This is detailed in Section 2 and Proposition 2.4, with relevant prior work cited. (ii) In the Gaussian variational family, exact LSVI can be tailored to eliminate the need to invert the Fisher information matrix. The resulting procedures have computational complexity $\mathcal{O}(d^3)$ in the full-covariance case and $\mathcal{O}(d)$ in the mean-field case, as shown in Section 4, specifically Theorems 4.1 and D.2. (iii) Under standard optimization assumptions, LSVI converges at explicit rates, established in Theorem 3.4 (Section 3). (iv) Empirical results demonstrate that LSVI achieves performance comparable to state-of-the-art variational inference methods and remains effective for non-differentiable target densities, as shown in Section 5 and Appendix C.

   Guidelines:

   - The answer NA means that the abstract and introduction do not include the claims made in the paper.
   - The abstract and/or introduction should clearly state the claims made, including the contributions made in the paper and important assumptions and limitations. A No or NA answer to this question will not be perceived well by the reviewers.
   - The claims made should match theoretical and experimental results, and reflect how much the results can be expected to generalize to other settings.
   - It is fine to include aspirational goals as motivation as long as it is clear that these goals are not attained by the paper.

2. **Limitations**

   Question: Does the paper discuss the limitations of the work performed by the authors?

   Answer: [Yes]

   Justification: (i) The approach is currently restricted to exponential families. Extending LSVI to more general variational families, such as mixtures of exponential families, is a promising direction for future work. (ii) Convergence guarantees for the stochastic versions of LSVI, namely, LSVI-MF and LSVI-FC, are not yet established. (ii) A comprehensive theoretical analysis of LSVI's convergence when the target distribution $\pi$ is replaced by an unbiased estimator $\hat{\pi}$ (e.g., via subsampling) remains an open problem. (iii) A comprehensive study of the constants involved in the convergence rates, in particular with respect to the smallest singular value of the FIM $r$, the latent dimension $d$ and the dimension of the statistic $m$ is left for future work. We believe most of the proofs can be adapted but the analysis would be more involved. See Section 6.

   Guidelines:

   - The answer NA means that the paper has no limitation while the answer No means that the paper has limitations, but those are not discussed in the paper.
   - The authors are encouraged to create a separate "Limitations" section in their paper.
   - The paper should point out any strong assumptions and how robust the results are to violations of these assumptions (e.g., independence assumptions, noiseless settings, model well-specification, asymptotic approximations only holding locally). The authors should reflect on how these assumptions might be violated in practice and what the implications would be.
   - The authors should reflect on the scope of the claims made, e.g., if the approach was only tested on a few datasets or with a few runs. In general, empirical results often depend on implicit assumptions, which should be articulated.

- The authors should reflect on the factors that influence the performance of the approach. For example, a facial recognition algorithm may perform poorly when image resolution is low or images are taken in low lighting. Or a speech-to-text system might not be used reliably to provide closed captions for online lectures because it fails to handle technical jargon.
- The authors should discuss the computational efficiency of the proposed algorithms and how they scale with dataset size.
- If applicable, the authors should discuss possible limitations of their approach to address problems of privacy and fairness.
- While the authors might fear that complete honesty about limitations might be used by reviewers as grounds for rejection, a worse outcome might be that reviewers discover limitations that aren't acknowledged in the paper. The authors should use their best judgment and recognize that individual actions in favor of transparency play an important role in developing norms that preserve the integrity of the community. Reviewers will be specifically instructed to not penalize honesty concerning limitations.

3. **Theory assumptions and proofs**

   Question: For each theoretical result, does the paper provide the full set of assumptions and a complete (and correct) proof?

   Answer: [Yes]

   Justification: All assumptions are provided in the core manuscript, see Theorem 2.1, Theorem 2.5, Theorem 3.1, Theorem 3.2 and Theorem 3.3. Furthermore, the assumptions are discussed in the core of the paper along with references mentioning existing similar assumptions in the VI literature. The proofs are deferred to the supplementary materials and are divided in several comprehensive steps, including lemmas in order to make the proof-reading procedure easier.

   Guidelines:

   - The answer NA means that the paper does not include theoretical results.
   - All the theorems, formulas, and proofs in the paper should be numbered and cross-referenced.
   - All assumptions should be clearly stated or referenced in the statement of any theorems.
   - The proofs can either appear in the main paper or the supplemental material, but if they appear in the supplemental material, the authors are encouraged to provide a short proof sketch to provide intuition.
   - Inversely, any informal proof provided in the core of the paper should be complemented by formal proofs provided in appendix or supplemental material.
   - Theorems and Lemmas that the proof relies upon should be properly referenced.

4. **Experimental result reproducibility**

   Question: Does the paper fully disclose all the information needed to reproduce the main experimental results of the paper to the extent that it affects the main claims and/or conclusions of the paper (regardless of whether the code and data are provided or not)?

   Answer: [Yes]

   Justification: Section 5 and Appendix C along with the pseudo-code Algorithms given in Section 3 are sufficient to reproduce all the experimental results. In particular, all input parameters are provided in Table 3.

   Guidelines:

   - The answer NA means that the paper does not include experiments.
   - If the paper includes experiments, a No answer to this question will not be perceived well by the reviewers: Making the paper reproducible is important, regardless of whether the code and data are provided or not.
   - If the contribution is a dataset and/or model, the authors should describe the steps taken to make their results reproducible or verifiable.

- Depending on the contribution, reproducibility can be accomplished in various ways. For example, if the contribution is a novel architecture, describing the architecture fully might suffice, or if the contribution is a specific model and empirical evaluation, it may be necessary to either make it possible for others to replicate the model with the same dataset, or provide access to the model. In general. releasing code and data is often one good way to accomplish this, but reproducibility can also be provided via detailed instructions for how to replicate the results, access to a hosted model (e.g., in the case of a large language model), releasing of a model checkpoint, or other means that are appropriate to the research performed.

- While NeurIPS does not require releasing code, the conference does require all submissions to provide some reasonable avenue for reproducibility, which may depend on the nature of the contribution. For example
  (a) If the contribution is primarily a new algorithm, the paper should make it clear how to reproduce that algorithm.
  (b) If the contribution is primarily a new model architecture, the paper should describe the architecture clearly and fully.
  (c) If the contribution is a new model (e.g., a large language model), then there should either be a way to access this model for reproducing the results or a way to reproduce the model (e.g., with an open-source dataset or instructions for how to construct the dataset).
  (d) We recognize that reproducibility may be tricky in some cases, in which case authors are welcome to describe the particular way they provide for reproducibility. In the case of closed-source models, it may be that access to the model is limited in some way (e.g., to registered users), but it should be possible for other researchers to have some path to reproducing or verifying the results.

5. **Open access to data and code**

Question: Does the paper provide open access to the data and code, with sufficient instructions to faithfully reproduce the main experimental results, as described in supplemental material?

Answer: [Yes]

Justification: The paper provides a Python (JAX) package that includes all discussed Algorithms (LSVI Algorithm 1, MF-LSVI Algorithm 2, FC-LSVI Algorithm 3, Variance control for the stepsizes Algorithm 4, NGD with details provided in Section C) as well as scripts to reproduce all the listed experiments. The package is explicitly divided into two parts, `variational` contains the generic implementations while `experiments` contains three sub-folders for the three distinct variational problems (logistic regression, variable selection and Bayesian synthetic likelihood). Full-pipeline for the experiments is provided (download and pre-processing of the datasets, inference procedures and post-processing scripts).

Guidelines:

- The answer NA means that paper does not include experiments requiring code.
- Please see the NeurIPS code and data submission guidelines (`https://nips.cc/public/guides/CodeSubmissionPolicy`) for more details.
- While we encourage the release of code and data, we understand that this might not be possible, so "No" is an acceptable answer. Papers cannot be rejected simply for not including code, unless this is central to the contribution (e.g., for a new open-source benchmark).
- The instructions should contain the exact command and environment needed to run to reproduce the results. See the NeurIPS code and data submission guidelines (`https://nips.cc/public/guides/CodeSubmissionPolicy`) for more details.
- The authors should provide instructions on data access and preparation, including how to access the raw data, preprocessed data, intermediate data, and generated data, etc.
- The authors should provide scripts to reproduce all experimental results for the new proposed method and baselines. If only a subset of experiments are reproducible, they should state which ones are omitted from the script and why.

- At submission time, to preserve anonymity, the authors should release anonymized versions (if applicable).
- Providing as much information as possible in supplemental material (appended to the paper) is recommended, but including URLs to data and code is permitted.

6. **Experimental setting/details**

Question: Does the paper specify all the training and test details (e.g., data splits, hyperparameters, how they were chosen, type of optimizer, etc.) necessary to understand the results?

Answer: [Yes]

Justification: The paper provides all the necessary details to reproduce the experiments, including the hyperparameters (the number of samples $N$, the number of iterations $T$, the initialisation distributions and the schedules) which are given in Section C. Different schedules have been considered to demonstrate robustness of the proposed methods while the number of samples is set to obtain reasonable numerical stability.

Guidelines:

- The answer NA means that the paper does not include experiments.
- The experimental setting should be presented in the core of the paper to a level of detail that is necessary to appreciate the results and make sense of them.
- The full details can be provided either with the code, in appendix, or as supplemental material.

7. **Experiment statistical significance**

Question: Does the paper report error bars suitably and correctly defined or other appropriate information about the statistical significance of the experiments?

Answer: [Yes]

Justification: All experiments were conducted using multiple trials as indicated in the figure labels and the Appendix C. For the logistic regression examples, one standard-deviation confidence intervals are provided over $100$ independent realisations. For the variable selection problem, the means and the min-max intervals for the posterior marginal probabilities obtained via LSVI over $100$ independent realisations. No statistical assumption is made for uncertainty measurement.

Guidelines:

- The answer NA means that the paper does not include experiments.
- The authors should answer "Yes" if the results are accompanied by error bars, confidence intervals, or statistical significance tests, at least for the experiments that support the main claims of the paper.
- The factors of variability that the error bars are capturing should be clearly stated (for example, train/test split, initialization, random drawing of some parameter, or overall run with given experimental conditions).
- The method for calculating the error bars should be explained (closed-form formula, call to a library function, bootstrap, etc.)
- The assumptions made should be given (e.g., Normally distributed errors).
- It should be clear whether the error bar is the standard deviation or the standard error of the mean.
- It is OK to report 1-sigma error bars, but one should state it. The authors should preferably report a 2-sigma error bar than state that they have a 96% CI, if the hypothesis of Normality of errors is not verified.
- For asymmetric distributions, the authors should be careful not to show in tables or figures symmetric error bars that would yield results that are out of range (e.g. negative error rates).
- If error bars are reported in tables or plots, The authors should explain in the text how they were calculated and reference the corresponding figures or tables in the text.

8. **Experiments compute resources**

Question: For each experiment, does the paper provide sufficient information on the computer resources (type of compute workers, memory, time of execution) needed to reproduce the experiments?

Answer: [Yes]

Justification: The full hardware and software specifications are provided in Appendix C, specifically in Table 1, along with Figures 4 and 5, which report experiments runtime and memory usage. All performance statistics are computed using independent realisations for improved robustness. In addition, scripts for measuring the runtime and memory usage of the algorithms can be found in the package (`/experiments/{...}/time.py`). All experiments were successfully performed and are reported in Section 5 and Appendix C. There is the exception of the applicability of ADVI (PyMC3, [6]) on the MNIST dataset, which is explicitly stated in Appendix C. Instead, ADVI as provided by Blackjax [7] was used as a replacement to PyMC3.

Guidelines:

- The answer NA means that the paper does not include experiments.
- The paper should indicate the type of compute workers CPU or GPU, internal cluster, or cloud provider, including relevant memory and storage.
- The paper should provide the amount of compute required for each of the individual experimental runs as well as estimate the total compute.
- The paper should disclose whether the full research project required more compute than the experiments reported in the paper (e.g., preliminary or failed experiments that didn't make it into the paper).

9. **Code of ethics**

Question: Does the research conducted in the paper conform, in every respect, with the NeurIPS Code of Ethics https://neurips.cc/public/EthicsGuidelines?

Answer: [Yes]

Justification: We carefully read through the NeurIPS Code of Ethics, and we see no violation of any guideline.

Guidelines:

- The answer NA means that the authors have not reviewed the NeurIPS Code of Ethics.
- If the authors answer No, they should explain the special circumstances that require a deviation from the Code of Ethics.
- The authors should make sure to preserve anonymity (e.g., if there is a special consideration due to laws or regulations in their jurisdiction).

10. **Broader impacts**

Question: Does the paper discuss both potential positive societal impacts and negative societal impacts of the work performed?

Answer: [NA]

Justification: The paper focuses on VI methods, emphasizing theoretical analysis and algorithmic implementability. As such, the work is foundational in nature and does not directly pertain to real-world applications or deployments. Given its abstract and theoretical scope, it does not present identifiable positive or negative societal impacts, including concerns related to fairness, privacy, or misuse.

Guidelines:

- The answer NA means that there is no societal impact of the work performed.
- If the authors answer NA or No, they should explain why their work has no societal impact or why the paper does not address societal impact.
- Examples of negative societal impacts include potential malicious or unintended uses (e.g., disinformation, generating fake profiles, surveillance), fairness considerations (e.g., deployment of technologies that could make decisions that unfairly impact specific groups), privacy considerations, and security considerations.

- The conference expects that many papers will be foundational research and not tied to particular applications, let alone deployments. However, if there is a direct path to any negative applications, the authors should point it out. For example, it is legitimate to point out that an improvement in the quality of generative models could be used to generate deepfakes for disinformation. On the other hand, it is not needed to point out that a generic algorithm for optimizing neural networks could enable people to train models that generate Deepfakes faster.
- The authors should consider possible harms that could arise when the technology is being used as intended and functioning correctly, harms that could arise when the technology is being used as intended but gives incorrect results, and harms following from (intentional or unintentional) misuse of the technology.
- If there are negative societal impacts, the authors could also discuss possible mitigation strategies (e.g., gated release of models, providing defenses in addition to attacks, mechanisms for monitoring misuse, mechanisms to monitor how a system learns from feedback over time, improving the efficiency and accessibility of ML).

11. **Safeguards**

Question: Does the paper describe safeguards that have been put in place for responsible release of data or models that have a high risk for misuse (e.g., pretrained language models, image generators, or scraped datasets)?

Answer: [NA]

Justification: We see no risk in the application of variational inference procedures.

Guidelines:

- The answer NA means that the paper poses no such risks.
- Released models that have a high risk for misuse or dual-use should be released with necessary safeguards to allow for controlled use of the model, for example by requiring that users adhere to usage guidelines or restrictions to access the model or implementing safety filters.
- Datasets that have been scraped from the Internet could pose safety risks. The authors should describe how they avoided releasing unsafe images.
- We recognize that providing effective safeguards is challenging, and many papers do not require this, but we encourage authors to take this into account and make a best faith effort.

12. **Licenses for existing assets**

Question: Are the creators or original owners of assets (e.g., code, data, models), used in the paper, properly credited and are the license and terms of use explicitly mentioned and properly respected?

Answer: [Yes]

Justification: All used Python packages are open-source, have permissive licenses, and are explicitly mentioned both in the manuscript and the code (`pyproject.toml` with complete dependency specifications). Specifically, Blackjax, PyMC, and JAX are mentioned in Section 1 and Section 5. Details on the datasets used, licenses and download links, are provided in Appendix C.

Guidelines:

- The answer NA means that the paper does not use existing assets.
- The authors should cite the original paper that produced the code package or dataset.
- The authors should state which version of the asset is used and, if possible, include a URL.
- The name of the license (e.g., CC-BY 4.0) should be included for each asset.
- For scraped data from a particular source (e.g., website), the copyright and terms of service of that source should be provided.
- If assets are released, the license, copyright information, and terms of use in the package should be provided. For popular datasets, `paperswithcode.com/datasets` has curated licenses for some datasets. Their licensing guide can help determine the license of a dataset.

- For existing datasets that are re-packaged, both the original license and the license of the derived asset (if it has changed) should be provided.
- If this information is not available online, the authors are encouraged to reach out to the asset's creators.

13. **New assets**

Question: Are new assets introduced in the paper well documented and is the documentation provided alongside the assets?

Answer: [Yes]

Justification: The provided Python (JAX) package for LSVI is well documented and includes a README file with instructions for installation and usage. The license is also included in the package (Apache License 2.0). In addition, we provide usage examples and accompanying commentaries.

Guidelines:

- The answer NA means that the paper does not release new assets.
- Researchers should communicate the details of the dataset/code/model as part of their submissions via structured templates. This includes details about training, license, limitations, etc.
- The paper should discuss whether and how consent was obtained from people whose asset is used.
- At submission time, remember to anonymize your assets (if applicable). You can either create an anonymized URL or include an anonymized zip file.

14. **Crowdsourcing and research with human subjects**

Question: For crowdsourcing experiments and research with human subjects, does the paper include the full text of instructions given to participants and screenshots, if applicable, as well as details about compensation (if any)?

Answer: [NA]

Justification: No experiment involving human subjects were conducted.

Guidelines:

- The answer NA means that the paper does not involve crowdsourcing nor research with human subjects.
- Including this information in the supplemental material is fine, but if the main contribution of the paper involves human subjects, then as much detail as possible should be included in the main paper.
- According to the NeurIPS Code of Ethics, workers involved in data collection, curation, or other labor should be paid at least the minimum wage in the country of the data collector.

15. **Institutional review board (IRB) approvals or equivalent for research with human subjects**

Question: Does the paper describe potential risks incurred by study participants, whether such risks were disclosed to the subjects, and whether Institutional Review Board (IRB) approvals (or an equivalent approval/review based on the requirements of your country or institution) were obtained?

Answer: [NA]

Justification: No experiment involving human subjects were conducted.

Guidelines:

- The answer NA means that the paper does not involve crowdsourcing nor research with human subjects.
- Depending on the country in which research is conducted, IRB approval (or equivalent) may be required for any human subjects research. If you obtained IRB approval, you should clearly state this in the paper.

- We recognize that the procedures for this may vary significantly between institutions and locations, and we expect authors to adhere to the NeurIPS Code of Ethics and the guidelines for their institution.
- For initial submissions, do not include any information that would break anonymity (if applicable), such as the institution conducting the review.

16. **Declaration of LLM usage**

Question: Does the paper describe the usage of LLMs if it is an important, original, or non-standard component of the core methods in this research? Note that if the LLM is used only for writing, editing, or formatting purposes and does not impact the core methodology, scientific rigorousness, or originality of the research, declaration is not required.

Answer: [NA]

Justification: There is no mention of LLMs in the manuscript, and no LLM was used.

Guidelines:

- The answer NA means that the core method development in this research does not involve LLMs as any important, original, or non-standard components.
- Please refer to our LLM policy (`https://neurips.cc/Conferences/2025/LLM`) for what should or should not be described.

