# OpenReview forum: "Least squares variational inference"
_NeurIPS.cc/2025/Conference — NeurIPS 2025 poster_

### Official Review · Reviewer_VdPC · 2025-06-28

**Clarity:** 4
**Significance:** 3
**Originality:** 3
**Rating:** 5
**Confidence:** 4

**Summary:**

We have a target distribution of interest, $\pi$. The goal is to design a VI method for exponential family variational distribution $q_\eta$ in a gradient free way, i.e., we do not make use of $\nabla \log \pi$. An unnormalised KL is introduced that shares the same optima as the standard KL. Then it is show that the stationary points of the unnormalised KL satisfy
$$
E_\eta [s s^T] \eta = E_\eta [f s]
$$
where $s$ is the score, $f=\log \pi$. The connection to least squares is immediate. An iterative algorithm is outlined.

For moderate-dimensional exponential families, we form monte carlo estimates of the $E_\eta$ above and directly invert the FIM. Some principled tricks are introduced in the case of high-dimensional Gaussian families that allows us to avoid this costly inversion.

**Questions:**

1. I believe the acronym MD, first used in Line 93 is never spelled out?
2. In Prop 2.2, you write $\eta=(\eta_0, \bar \eta)$. Then in Algorithm 1, the notation $\eta_0$ is double-loaded?
3. The cost of inverting $F$ is mitigated only in the case of Gaussians. What about other exponential families where the computation time could still be formidable? Is this, at the moment, only a practical methodology for Gaussian variational families?
4. Aren't there other gradient-free VI methods out there? Should these be discussed and used as empirical baselines?

**Ethical Concerns:**

["NO or VERY MINOR ethics concerns only"]

**Final Justification:**

I maintain my very positive score. I disagree with the reviewer that gave a 3 who felt the paper was not well written or presented. This paper is very clear mathematically, far surpassing the typical ML conference submission. The same reviewer also found a lack of novelty. This is subjective of course but I disagree. Often with ML papers, the novelty is front-loaded and so much time is spent on narrative, and so little on the actual content. This paper is very clear in what it is doing, I understand what is new and what is not new just from the writing. I think what is new is sufficient novelty.

**Limitations:**

Yes

**Quality:**

4

**Strengths And Weaknesses:**

**Strengths**

* The paper is technically solid and self-contained.
* While the scope is restricted to exponential-family variational approximations, this focus is a strength, not a liability: it yields clean mathematics and clear algorithms.
* Even ignoring the formal theory, the fixed-point (OLS) view and the Fisher-inversion-free Gaussian trick are valuable practical contributions to variational inference.

**Weaknesses**

* Experimental evidence is limited to small-scale benchmarks; larger or more diverse tasks would better demonstrate practical impact.
* It remains unclear how often exponential-family variational forms are the best practical choice in modern applications—guidance or examples here would strengthen the paper.

---

> ### Author Rebuttal · Authors · 2025-07-28
>
> We thank the reviewer for his feedback and supportive comments. Below, we address the raised points.
>
> 1. We agree that higher-dimensional benchmarks for our methods would further
>    strengthen the paper. Regarding the variety of the inference tasks: we
>    provide numerical examples on both discrete and continuous target
>    distributions, including case where the likelihood is intractable.  Another
>    potential example, where gradient-based procedures are not implementable, is
>    PAC-Bayes learning, in which the target distribution takes the form
>    $\pi(\theta)\propto \mathcal{N}(\theta| \theta_0, C) e^{-\lambda
>    \hat{R}(\theta)}$, for some $\lambda > 0$, and where $\hat{R}$ is a empirical
>    risk function (e.g., the misclassification rate in a classification task). Indeed, since $\hat{R}$ is typically piecewise constant,
>    its gradient is almost-everywhere zero.
>
> 2. On the restriction to exponential families, we refer the reviewer to our
>    answer to Reviewer aZSa on the same point.
>
> **Questions**:
>
> 1. We now spell out "Mirror Descent (MD)" the first time it occurs, thank you for catching this.
> 2. We agree, and now use the notation $\eta = (\eta^{(0)}, \bar{\eta})$ in Proposition 2.2.
> 3. When the variational family is the independent product of Bernoulli distributions indexed by
>    $p=(p_1, \ldots, p_d)^{\top}$, the structure of the FIM can be leveraged to propose a low-complexity scheme.
>    Indeed, in that case, $F$ is the sum of a diagonal matrix $D=-\textup{diag}(p)$, and a rank-one
>    matrix $pp^{\top} =(p_ip_j)$, and
>    therefore, the FIM can be inverted with little cost via Woodbury's matrix identity.
>    (This is something we have realised only recently, and have not yet
>    implemented.)
> 4. To the best of our knowledge, only Arenz et al. (2018) propose a gradient-free method for VI.
>    We do mention it in the logistic regression example (Section 5.1), and compare it to our methods.
>
> **References**
>
> - Arenz O., Neumann G., and Zhong M., Efficient gradient-free variational inference using policy search, PMLR, 2018.

---

### Official Review · Reviewer_aZSa · 2025-07-01

**Clarity:** 2
**Significance:** 3
**Originality:** 2
**Rating:** 4
**Confidence:** 3

**Summary:**

This paper introduces Least Squares Variational Inference (LSVI), a novel gradient-free method for variational inference in exponential families. The core idea is to reformulate the variational fixed-point equation as a sequence of least squares regressions on tempered log-target evaluations. The authors show LSVI corresponds to a biased stochastic natural gradient descent and provide convergence guarantees under mild assumptions. They also develop efficient algorithms for Gaussian variational families, avoiding Fisher matrix inversion and enabling linear or cubic complexity depending on the covariance structure. The method is validated on tasks including logistic regression, discrete variable selection, and likelihood-free inference, demonstrating strong performance even when gradients are unavailable.

**Questions:**

1. Could LSVI be extended to handle mixtures or flows beyond exponential families?
2. How stable is the method in higher-dimensional settings where Monte Carlo estimates of the FIM become noisy?
3. Could the authors clarify under what conditions the regression operator stays within the variational family?
4. How does LSVI compare in practice with Black-Box VI methods on models with intractable likelihoods?

**Ethical Concerns:**

["NO or VERY MINOR ethics concerns only"]

**Final Justification:**

The comments by the authors have clarified some of my doubts, so I am more confident in the original evaluation of the paper.

**Limitations:**

- Currently restricted to exponential-family approximations.
- No discussion on combining LSVI with amortized inference or VI for deep generative models.
- Adaptive step size scheme is proposed but the method is quite sensitive to this point. A sensitivity and ablation study could be useful here.
- The computational gain over NGD could be better quantified in real high-dimensional applications.

**Quality:**

3

**Strengths And Weaknesses:**

**Strengths**
- Proposes a technically novel and practically useful gradient-free VI method that works even in non-differentiable or discrete settings.
- Theoretical analysis is solid, with convergence guarantees for both the exact and stochastic schemes under reasonable assumptions.
- Offers efficient, tailored implementations for Gaussian families, with competitive time complexity.
- Demonstrates flexibility across a range of problems including variable selection and Bayesian synthetic likelihood.
- Writing is rigorous, with many technical details included in the appendices. Notation is generally clear, though some could be improved for clarity.

**Weaknesses**
- While well-motivated, the method is limited to exponential families, which constrains broader applicability.
- Performance comparisons against deep VI models or more flexible approximation families are missing.
- The connection to prior works, such as stochastic linear regression VI (e.g. Salimans & Knowles, 2013) could be more explicitly elaborated.
- The choice of referring to the appendix to clarify the notation is not a good idea, as it makes the main text harder to follow. I suggest the authors make the notation clearer in the main text, or at least provide a summary of the notation there.
- The practical choice of step size remains somewhat heuristic; adaptive methods are proposed but not systematically evaluated.

**Minor**
- The GitHub link could be made more visible, moved to a footnote or explicitly listed in the conclusion for better visibility.

---

> ### Author Rebuttal · Authors · 2025-07-28
>
> Many thanks for your feedback. We address below the mentioned weaknesses.
>
> 1. We agree that our method is currently limited to exponential families. Here are some ways of
>   lifting this restriction:
>
> - Mixtures of exponential families may be tackled by adapting the Expectation Maximimisation approach of
>   [Arenz et al., 2013].
> - For Gaussian approximations, if the posterior contains directions that are strongly non-Gaussian, then
>   conditional-Gaussian strategies like INLA [Rue et al., 2009] may be applied.
> - Another way to improve a Gaussian approximation is to use the skew-symmetric modification recently proposed
>   by [Pozza et al., 2024].
> - In discrete exponential families, independence can be lifted by considering tree-structured
>   dependencies [Wainwright & Jordan, 2008], which are quite flexible.
>
>   We will add this discussion to the conclusion.
>
> 2. However, we currently do not have a way of directly using LSVI to fit a family of densities defined by deep models via
>    normalising flows. They can perhaps be fitted using layer-wise linearisation, but this is at this stage
>    just a direction for future research.
> 3. Thank you for your remark, we agree, and we will elaborate on this in the final version of this manuscript.
> 4. Thank you for this suggestion. We moved our notation table to the main text.
> 5. Choosing the stepsizes for optimisation procedures in VI is recognised as a challenging problem [Welandawe et al., 2024]. We
>    do provide in the Appendix a comparison of the KL losses using different strategies for choosing the learning step,
>    including, adaptive rule (Alg. 4), and fixed step sizes, see Table 3, and Figures 5, 6, 7 in Appendix D. Any other
>    stepsize strategy could be adapted to our method, such as line search to ensure better convergence results,
>    however, at the cost of additional loss evaluations.  Additionally, as the proposed algorithms are stochastic, a
>    good strategy to stabilise the results is to average over the last iterations.
>
> **Questions**:
>
> 1. The expectation-maximisation approach of Arenz, et al. could be extended to handle mixtures of exponential families.
> 2. While a rigorous analysis of the convergence of LSVI with a control on the dimension $d$ is out of reach for now, we
>    found the sample complexities for the Fisher matrix in the mean-field case to be $O(d\log(d)^2)$, and in the
>    full-covariance case $O(d\log(d)^2)$.
>    Therefore, to maintain stable numerical results as the dimension increases, one needs to increase the number of
>    samples accordingly.
> 3. When the target distribution $\pi$ is in the variational family, the regression operator recovers the exact natural
>    parameter, in one iteration.
>    However, in general, the regression operator is not in the natural space, and therefore, we consider a momentum
>    iteration (Eq. 11), by taking a sufficiently small stepsize, which is always possible since the variational family is
>    an open set under Assumption 2.1.
>
> **References**
>
> - Arenz O., Neumann G., and Zhong M., Efficient gradient-free variational inference using policy search, PMLR, 2018.
> - Rue H., Martino S., and Chopin N., Approximate Bayesian inference for latent Gaussian models by using integrated
>   nested Laplace approximations, JRSSB, 2009.
> - Pozza F., Durante D. and Szabo B., Skew-symmetric approximations of posterior distributions, arXiv:2409.14167, 2024.
> - Martin J. Wainwright and Michael I. Jordan (2008), Graphical Models, Exponential Families, and Variational
>   Inference, Foundations and Trends® in Machine Learning: Vol. 1.
> - Welandawe M., Andersen M., Vehtari A., and Huggins J., A framework for improving the reliability of black-box
>   variational inference. Journal of Machine Learning Research.

---

> > ### Comment · Reviewer_aZSa · 2025-08-07
> > **Response to the authors**
> >
> > Thank you for the rebuttal. I appreciate the thoughtful answers and the willingness to address the points raised.
> >
> > - The current focus on exponential families is understandable. The new discussion in the conclusion, outlining possible extensions via EM methods, INLA-style strategies, or recent skew-symmetric corrections is helpful. Mentioning tree-structured models for discrete cases also adds useful context.
> >
> > - The comment on normalizing flows is fair—while not immediately compatible with LSVI, the idea of layer-wise linearization is an interesting research direction worth briefly mentioning as future work.
> >
> > - Moving the notation table to the main text is a good decision. It improves readability and avoids unnecessary back-and-forth for the reader.
> >
> > - The clarification on step size is appreciated. The results in the appendix give a clearer picture of different strategies, and the suggestion to average final iterates is practical. Including this more explicitly in the main text would be beneficial.
> >
> > - Regarding higher-dimensional stability, the empirical scaling of sample complexity—linear for mean-field, cubic for full-covariance—makes sense. This point is worth highlighting to guide practitioners.
> >
> > - The explanation about convergence and the role of the regression operator is clear. Emphasizing that exact recovery holds only when the true posterior lies in the variational family, and that otherwise convergence is handled via damped updates, is important.
> >
> > Overall, these clarifications strengthen the paper. I encourage the authors to integrate them in the final version to better support readers and position LSVI within the broader VI landscape.

---

### Official Review · Reviewer_WRbV · 2025-07-02

**Clarity:** 2
**Significance:** 2
**Originality:** 3
**Rating:** 4
**Confidence:** 2

**Summary:**

In this work, the authors propose a gradient-free and Monte Carlo-based scheme version of the vanilla variational autoencoder, named the  Least Squares Variational Inference (LSVI). Remarkably, this framework proposes a gradient-free variational inference procedure that reframes the traditional KL divergence minimization target as a sequence of ordinary least squares (OLS) regression problems. This method enables optimization in exponential-family distributions that does not requiring gradients of the log-target density. This framework can be applied to various tasks, like the logistic regression, discrete variable selection, and Bayesian synthetic likelihood.

**Questions:**

1. Could you provide also a literature review of the paper in this VAE with more theoretical guarantees domain? Like, [1], [2].
2. Could you apply this method to any real-word datasets and get the experiment results?




[1] Least square variational Bayesian autoencoder with regularization. Ramachandra, et, al. 2017.

[2] Forward \chi^2 Divergence Based Variational Importance Sampling. Li, et, al. 2023.

**Ethical Concerns:**

["NO or VERY MINOR ethics concerns only"]

**Final Justification:**

See my full comments in the review and below. I thank the authors for the clarifications and I will keep my rating at 4.

**Limitations:**

Please refer to my 'Strengths And Weaknesses' and 'Questions' section.

**Quality:**

3

**Strengths And Weaknesses:**

As for the strengthes of the method: The math part of the papers are rigorous and well-written. It's clear to me and the Propositions and the exact LSVI scheme Section 2.1 makes sense to me. Moreover, this work connect the LSVI theoretically to normal gradient descent and mirror descent, and shows the proved convergence rates under mild assumptions. Empirical results on the numerical experiments are widely conducted on the logistic regression, discrete variable selection, and Bayesian synthetic likelihood tasks, which I think is solid.

For the weaknesses, I think that in the Introduction section, the motivation of the work is not too clear and also the Eq. (1) does not look like the traditional VAE formula showed in its original paper [1]. Moreover, as the VAE framework is proposed for introducing the power of deep neural networks into the variational bayes framework, then adding the constraints listed in this work is actually a trade-off and the impact of this idea would be a bit limited. Thirdly, the method tends to provide the point estimates rather than with UQ.

[1] Auto-Encoding Variational Bayes. Kingma, et, al. 2014.

---

> ### Author Rebuttal · Authors · 2025-07-28
>
> We are happy to cite more extensively the VAE litterature, including the paper
> you mentioned. Please note however that our focus is on VB (Variational Bayes),
> that is, a parametric approximation of a *fixed* posterior distribution, which
> is a bit different from VAEs. In a VAE, the variational approximation is only
> one component of the whole procedure, which also comprises maximisation with
> respect to both $\theta$ (a model hyper-parameter) and $\phi$ (parameter of the
> approximation family), using the notations of Kingma and Welling (2014). In our
> case, we do not have a $\theta$. In addition it does not look straightforward
> to plug neural networks in our approach, as done in VAE (although it may be an
> interesting line of research in the future). In fact, the neural networks in
> VAE are introduced to map an input  $x$ to a Gaussian approximation of
> $p(z|x)$, based on a large sample of $x_i$ (hence one has as many targets as
> there are $x_i$), but again in our case we have a single target.

---

> > ### Comment · Reviewer_WRbV · 2025-08-06
> >
> > I thank the authors for the clarification to underscore the focus on VB.

---

### Official Review · Reviewer_7qp3 · 2025-07-03

**Clarity:** 3
**Significance:** 4
**Originality:** 4
**Rating:** 5
**Confidence:** 3

**Summary:**

This work proposes a method of iteratively running least squares regressions to fit the parameters for variational inference over exponential families. In particular, the special case of fitting an exponential variational family can be shown to be equivalent to solving a fixed point problem. With this starting point, the authors of this paper then proceeded to demonstrate how such a fixed point method could be solved, first exactly assuming oracle access to expectations and then with practical Monte Carlo approximations. They further demonstrated that this MC estimate has convergence guarantees on the recovery of the true optimal variational parameters as would be produced under exact LSVI. Even this practical MC algorithm, however, involves an expensive step of inverting the Fisher Information Matrix (FIM), rendering this procedure more costly than would likely be desirable. In particular cases, specifically concentrating on the Gaussian case in this paper, such an inversion can be circumvented. Finally, the method was robustly demonstrated to perform well across many empirical settings.

**Questions:**

1. Mostly out of curiousity, where did the inspiration of introducing t(z) to compute the FIM inverse emerge from? Is this common in some other subfield?

**Ethical Concerns:**

["NO or VERY MINOR ethics concerns only"]

**Final Justification:**

I maintain my original stance and believe this work is a technically sound and interesting contribution to the VI space. I, therefore, stand by my original score for acceptance.

**Limitations:**

Yes

**Quality:**

4

**Strengths And Weaknesses:**

-------
Quality
-------
There were no technical flaws in either the theorem statements or proofs that I was able to identify. The proof assumptions were all stated precisely and explicitly referenced in the statements of the theorems. Additionally, the empirical findings were properly framed against alternate variational inference estimation techniques (i.e. in Figures 1 and 2).

-------
Clarity
-------
The paper was very clearly presented. As stated in the previous discussion, all the theorem statements were precisely stated and the assumptions similarly explicitly provided. There were a few spots that would have benefitted from some exposition to give clarity on significance, specifically on the convergence guarantees of Theorem 3.4, where the decomposition of the three terms (i.e. first term for initialization error, second for Monte Carlo error, and third for iteration convergence) that would have made parsing the significance of some of the results and where they arose from from the proofs more direct. Nonetheless, the authors provided comprehensive discussions around the difference between the provided results in the broader context of other works in this space, such as in the discussion of Remark 3.5, which made identifying contributions of the work quite immediate.

In addition to the individual theorem statements, the flow of the paper was also very well done, where complexities were gradually layered into the final formulation, starting with the exact solution to the fixed point problem, followed by a relaxation to the exactness of the expectations, and finally to the special case of Gaussians. This made following the progression of complexity much easier to follow along. Additionally, the text in the main body was further well-encapsulated in the algorithms, which also used consistent notation as the main body, making it straightforward to follow along with. Finally, the plots and presented results of the experiments section were also cleanly presented and easy to interpret, even in isolation of the main text.

------------
Significance
------------
This paper contributes significantly to the area of variational inference, both in likelihood-based and likelihood-free settings. As discussed, the method proposed is ultimately most useful when fitting Gaussian variational approximations due to the tractability of estimating the FIM inverse in the manner outlined in the paper. Nonethless, there are many contributions that all seem very worthwhile. The first is simply the insight that variational inference can be computed in this iterative least squares fashion: this is a very elegant and much more generally applicable insight than just in the special case of Gaussianity. Other researchers can, therefore, leverage this reframing for the development of similarly tractable estimation strategies to layer atop this initial insight of reframing. Additionally, the estimation of the inverse FIM matrix for Gaussian families was a very elegant trick to allow for the reduction from the naive $O(d^6)$ cost to $O(d^3)$ via backsubstitution. It is not exactly clear to me if this type trick has been done in other literature, but I have not seen it referenced in the space of the fitting of variational families, which alone makes its presentation to this community a worthwhile contribution.

-----------
Originality
-----------
As discussed above, this paper provides many novel insights, both in the original framing of the variational inference problem as a regression problem and in more granular technical details around the FIM inversion (i.e. in the reparameterization with Eqns 16 and 17). These are novel and worthwhile contributions to the variational inference and also likelihood-free inference communities, whether or not such insights were present in orthogonal communities of statistics.

---

> ### Author Rebuttal · Authors · 2025-07-28
>
> Many thanks for your supportive comments and constructive feedback.
>
> Regarding the readability of the theoretical results, in particular the
> convergence guarantees Theorem 4.1, we made it easier for the reader to parse
> and understand the practical consequences of this result (i.e., convergence in
> average of the average of the iterates). We also added a remark on the
> decomposition of the convergence error (i.e., deterministic error due to
> mirror and the initilisation, mirror descent error coupled to Monte Carlo error, and the vanishing Monte Carlo
> bias). We also added explanations between the steps of the proofs to
> enhance fluidity of the manuscript.
>
> **Questions**:
> The intuition for the reparameterization comes from nonparametric regression,
> i.e. recovering a function $f$ from data $(X_i, Y_i)$ such that $Y_i = f(X_i) +
> \epsilon_i$, for some zero-mean i.i.d random variables $\epsilon_i$. A common strategy is to decompose $f$ on a functional orthonormal basis.
> Then the predictors (the elements of the basis) are independent, and it is
> possible to perform $p$ independent linear regressions (based on a single
> predictor) instead of one (based on all the predictors). In our case, the
> statistic $t$ is constructed as the normalised Hermite polynomials of order at
> most two, which form an orthogonal basis for the vector space of polynomials of
> order two under the Gaussian measure.

---

### Official Review · Reviewer_X8ht · 2025-07-03

**Clarity:** 2
**Significance:** 2
**Originality:** 2
**Rating:** 3
**Confidence:** 3

**Summary:**

This paper proposes an new variational inference method called least square variational inference.
A connection has been drawn to natural gradient descent and mirror descent, which is used to prove a convergence rate of the algorithm.
Experiments demonstrate marginal improvement against natural gradient descent.

**Questions:**

1. In the last line Algorithm 3, why there is an "unvec"?
If I understand correctly, \\(\hat\eta_{2, t + 1}\\) is already a matrix.
So I am not sure what "unvec" does here.

1. Can we implement Eq (11) without explicitly inverting the Fisher information matrix for non-Gaussian variational distributions?
This is possible in natural gradient descent.

**Ethical Concerns:**

["NO or VERY MINOR ethics concerns only"]

**Final Justification:**

I initialled gave a 2. In the rebuttal, the author has cleared some of my confusions, e.g., why do we estimate the Fisher information matrix stochastically even though a closed form exists. However, it also strengthens my believe that the proposed method ultimately boils down to an (perhaps) improved stochastic gradient estimator for stochastic natural gradient descent. I'll willing to raise my score to borderline reject. But I still think there are two major weaknesses remaining: 1) the author didn't do a good job in presentation; b) the novelty is limited.

**Paper Formatting Concerns:**

No issues.

**Quality:**

2

**Strengths And Weaknesses:**

1. The writing is not good.
The paper is notation dense (which is fine), but the authors did not do a good job walking readers through the material.
    1. The matrix \\(F_\eta\\) is the Fisher information matrix, but this is never mentioned in the paper.
    1. I am very confused about Section 4 from Line 195 to Line 220.
    I think this section is intended to derive an efficient implementation for Eq (11) in the special case of Gaussian variational distributions.
    However, why does Theorem 4.1 have the name "exact LSVI mapping \\(\phi\\) for full-covariance Gaussian distributions"?
    If you have an exact form of \\(\phi\\), then you are able to implement the fixed-point equation  (6) with zero Monte Carlo error.
    Due to the connection between LSVI and NGD, this implies that natural gradient descent can be implemented with zero Monte Carlo error even if the likelihood is non-conjugate??
    1. Not sure what \\(t(z)\\) is in Eq (16).

1. The theory is weak and uninteresting.
I feel like Assumption 2.5 basically does most of heavy lifting---it is unclear whether real VI objectives in practice are smooth or strongly convex.
In fact, there is evidence that the VI objective is not even convex for non-conjugate likelihoods, let alone strongly convex (Wu et al., 2024).

1. In Eq (10), why do we use Monte Carlo samples to approximate the Fisher information matrix? The Fisher information matrix is already available in a closed-form for (at least many of) exponential families.

1. Ultimately, Algorithm 3 is quite similar to natural gradient descent.
Although the authors present the paper as a new method LSVI, I think the proposed method actually boils down to an (perhaps) improved stochastic gradient estimator for stochastic natural gradient descent.
This is just my gut feeling though.

Wu, K., & Gardner, J. R. (2024). Understanding stochastic natural gradient variational inference. arXiv preprint arXiv:2406.01870.

---

> ### Author Rebuttal · Authors · 2025-07-28
>
> We sincerely thank the reviewer for their feedback, which has helped us clarify and improve the paper. Below, we address
> each point in detail.
>
> ### 1.1 Clarification on the Fisher information matrix (FIM)
>
> Agreed, we now mention that $F_\eta$ is the Fisher information matrix as soon
> as it is defined.
>
> ### 1.2. Exact LSVI for Gaussian Variational Families
>
> The term "exact" in "Exact LSVI Mapping for Full-Covariance Gaussian Distributions" (Th. 4.1) refers to
> the *oracle setting* where expectations (e.g., $ \mathbb{E}_q[\cdot] $) are computed exactly, as introduced in Section 2.
> This does *not* imply zero Monte Carlo error in practice. More precisely, Theorem 4.1 gives the LSVI mapping in term
> of $\gamma$ (Line 210), which is an intractable expectation (in most cases), and needs to be estimated via Monte Carlo.
> We will clarify this distinction in the text to avoid confusion.
>
> Regarding the connection to NGD: Indeed, LSVI and NGD coincide in the exact setting (with oracle expectations, Proposition
> 2.4). Our key contribution is a *gradient-free* stochastic approximation that avoids evaluating gradients of $ \log \pi $,
> plus matrix inversion of the FIM when the family is Gaussian (Algorithms 2, 3) .
>
> ### 1.3. Reparameterization and the role of $t(Z)$ in Eq. (16)
>
> In a nutshell, the point of $t(Z)$ is to define a simple, one-to-one
> transformation such that the output vector has un-correlated components:
> $\mathbb{E}[t(Z) t^\top(Z)] = I$. That makes possible to perform linear
> regression without inverting the FIM.
>
> We explicitly give the mapping from $\gamma$ to $\eta$ depending on $(\mu,
> \Sigma)$ (Theorem 4.1), and we derive the corresponding tailored stochastic
> schemes in 4.1. We now more thoroughly emphasize the role of $t$ in the text.
>
>
> ### 2. Convexity assumption
>
> We agree the strong-convexity/convexity conditions are rarely verified in
> practice, however, such assumptions are standard for analyzing convergence of
> optimization algorithms (including SGD and NGD), to ensure a unique minimizer
> and tractable rates (see Remark 2.6 and, e.g. Chap. 5 in the book of Bach,
> 2024).   While non-conjugate VI objectives may not be globally convex, *local*
> convexity near optima often suffices in practice. We emphasize that our theory
> provides guarantees under idealized conditions (as is common in the optimization
> literature), and empirical results demonstrate robustness beyond these
> settings. We will further clarify that:
>
> - Convergence guarantees for mirror descent can be derived under non-strongly convex (but convex) assumption (see Lines
>   115-120).
> - Local convexity near optima often suffices in practice, and local convergence to local minimisers still hold in that
>   case.
> - Convexity holds when the variational family contains the target (e.g., conjugate models, Wu et al., 2024).
>
> ### 3. Monte Carlo approximation of the FIM (Eq. 10) and variance reduction
>
> We agree that closed-form FIMs exist for several exponential families, however we explicitly use a Monte Carlo estimate of
> the FIM rather that its close-form expression.
> This is because our method treats the KL minimisation problem *as a linear regression task*: the Monte Carlo
> estimator $ \hat F_{\eta} $, (Eq. 10), is the *empirical FIM matrix*, and $ \hat z_{\eta} $ is the empirical moment
> vector (Eq. 10). As a consequence, the variance-optimal estimator for the least-squares problem is $ \hat F_{\eta} \hat z_
> {\eta} $, and not $ F_{\eta} \hat z_{\eta} $ where $ F_{\eta} $ is the closed-form FIM (i.e., no Monte Carlo error).
> To sum up:
>
> - Using the exact FIM $ F_{\eta} $ and the Monte Carlo moment vector $ \hat z_{\eta} $ to compute the regressor is not
>   optimal with respect to the variance.
> - Using the same samples for $ \hat F_{\eta} $ and $ \hat z_{\eta} $ reduces variance (as in standard linear
>   regression, see Remark 3.6).
>
> In the Gaussian case, we implicitly compute the exact inverse FIM via the reparameterization introduced in Section 4,
> specifically, Theorem 4.1. The resulting Algorithms 2, 3, are faster (i.e., smaller CPU cost per iteration), but noisier.
>
> ### 4. Relationship to natural gradient descent (NGD)
>
> We agree with you, as we state in Section 2.1, exact LSVI recovers NGD on the *unormalised* KL objective. Our
> focus is on *practical, gradient-free approximations* via a linear regression formulation:
>
> - Algorithm 1 does not require computing gradients of the log-target distribution, contrary to NGD, which enables us to
> tackle
> previously inaccessible models.
> - Our algorithms yield very stable numerical results, and are easy to implement, and to tune.
> - All our algorithms can leverage highly optimized routines for performing the linear regression.
>
> **Questions**:
>
> 1.`unvec` in Algorithm 3: The natural parameter $ \eta $ is a vector for general exponential families (i.e., the variational distribution is $
> q_\eta(x)\propto\exp(\eta^\top s(x)) $). For Gaussians, $ \eta_2 = -\frac{1}{2}\textup{vec}(\Sigma^{-1}) $ (Line 200), and the `unvec` reshapes the
> vector into the new precision matrix. Sorry for the confusion, we now clarify this.
>
> 2.Avoiding FIM Inversion for Non-Gaussian Families: For general exponential families, FIM inversion can only be avoided
> if gradients with respect to the moment parameters $ \omega $ are available (since $ \nabla_\omega \ell = F_
> \eta^{-1} \nabla_\eta \ell $). Without such gradients, explicit inversion is required.
>
> **References**
>
> - Bach. Learning Theory from First Principles, MIT Press, 2024.
> - Kaiwen Wu and Jacob R. Gardner. Understanding stochastic natural gradient variational inference, ICML 2024.

---

> ### Comment · Reviewer_X8ht · 2025-08-04
>
> > Our key contribution is a gradient-free stochastic approximation that avoids evaluating gradients of $ \log \pi $, plus matrix inversion of the FIM when the family is Gaussian (Algorithms 2, 3).
>
> I am trying to understand the gradient-free claim a bit better. Does the update rule on \\(\hat{z} _ {\hat\eta_t}\\) implicitly use the score function gradient estimator (as opposed to the common reparameterization trick gradient)? Is that why it's gradient-free? Or put it another way: It doesn't rely on automatic differentiation to compute the gradient because the score function gradient has a simple closed-form.
>
> > 3. Monte Carlo approximation of the FIM (Eq. 10) and variance reduction
>
> After reading this explanation, I think I understand the motivation a bit better.

---

> > ### Author Response · Authors · 2025-08-05
> > **Response by Authors**
> >
> > We agree with your insight: the gradient-free aspect of LSVI can be recovered by using the closed-form gradient of expectations under $q_{\eta}$ (via the log-derivative trick). In particular, since the exact update rule is equivalent to NGD on the unnormalised KL objective (for a specific choice of step sizes, Proposition 2.4), we can apply the log-derivative trick to simplify the gradient expression of the uKL (Eq. 28 in Appendix). In particular $\nabla\_\eta \mathbb{E}\_\eta [f(X)] = \mathbb{E}\_\eta [s(X)f(X)] = z\_\eta$, and ultimately $\hat z_\hat\eta$ can be thought of as a score function gradient estimator.
> >
> > In current VI practice, the need for gradients of $\log \pi$ arises from the use of the reparametrisation trick, which is the best known way to make stochastic-gradient VI stable enough for practical use.
> >
> > In our paper, the focus is on finding practical update steps, that are stable without the need for gradients (of $\log \pi$ or of expectations under $q_\eta$), and on studying their theoretical behaviour.

---

### Note · Authors · 2025-08-12

We thank all reviewers for their comments, which helped us to improve the clarity of the manuscript. All the mentioned modifications and additions will be included in the final manuscript.

---

### Decision · Program_Chairs · 2025-09-17

**Decision:**

Accept (poster)

**Comment:**

This paper considers the setting of variational inference using exponential family distributions as the approximating family. They consider the problem of minimizing the KL divergence over this family, which they show becomes a least squares problem. A key advantage of this method is that it does not require differentiability of the target, which is often required in many black-box VI formulations.  The paper shows that the LSVI procedure is a biased stochastic natural gradient descent procedure, and use this equivalence to derive rates of convergence. The method is demonstrated empirically on several examples, including a likelihood free problem.

The majority of reviewers were in favor of accepting this paper. One reviewer, who initially was in favor of rejection, raised their score recommendation after the rebuttal phase, stating that the discussion helped clear up some misconceptions regarding the motivation behind the Fisher information matrix. The reviewer felt that the paper overall could improve in clarity, e.g. defining notation more explicitly, and other reviewers had clarifying questions that, while addressed in the discussion, should be incorporated into a future revision. To address concerns about novelty, the reviewers suggest providing a deeper discussion of context and related work, e.g., emphasizing / clarifying the gradient-free contribution when discussing both the connection with NGD and to expand the discussion of the broader VI literature, which has primarily been focused on the case where the scores of the target are available.

I recommend accepting this paper, provided the authors address the remaining reviewers' comments.